# VAR-MATH: Probing True Mathematical Reasoning in LLMs via Symbolic Multi-Instance Benchmarks

## Abstract

Recent advances in reinforcement learning (RL) have led to substantial improvements in the mathematical reasoning abilities of large language models (LLMs), as measured by standard benchmarks. Yet these gains often persist even when models are trained with flawed signals, such as random or inverted rewards. This raises a fundamental question: do such improvements reflect genuine reasoning, or are they merely artifacts of overfitting to benchmark-specific patterns? To answer this question, we adopt an evaluation-centric perspective and highlight two critical shortcomings in existing protocols. First, *benchmark contamination* arises because test problems are publicly available, thereby increasing the risk of data leakage. Second, *evaluation fragility* results from reliance on single-instance assessments, which are sensitive to stochastic outputs and fail to capture reasoning consistency. These limitations suggest the need for a new evaluation paradigm that can probe reasoning ability beyond memorization and one-off success. As response, we propose **VAR-MATH**, a symbolic evaluation framework that converts fixed numerical problems into parameterized templates and requires models to solve multiple instantiations of each. This design enforces consistency across structurally equivalent variants, mitigates contamination, and enhances robustness through bootstrapped metrics. We apply VAR-MATH to transform three popular benchmarks, AMC23, AIME24, and AIME25, into their symbolic counterparts, VAR-AMC23, VAR-AIME24, and VAR-AIME25. Experimental results show substantial performance drops for RL-trained models on these variabilized benchmarks, especially for smaller models, with average declines of 47.9% on AMC23, 58.8% on AIME24, and 72.9% on AIME25. These findings indicate that some existing RL methods rely on superficial heuristics and fail to generalize beyond specific numerical forms.

## 1 Introduction

Recent advances in large language models (LLMs) have led to remarkable improvements in mathematical reasoning tasks. Models such as OpenAI-o1 (OpenAI, 2024), DeepSeek-R1 (Guo et al., 2025), and Kimi-k1.5 (Team et al., 2025) have achieved state-of-the-art results across a range of public benchmarks. A key contributor to this progress is the growing shift from conventional supervised fine-tuning (SFT) to reinforcement learning (RL), which has become a dominant strategy for aligning model outputs with desired reasoning behaviors. The impressive performance of models like DeepSeek-R1 has sparked a surge of research, which generally follows two directions. One focuses on improving data quality through filtering, deduplication, and verification pipelines (Meng et al., 2023; He et al., 2025b; Hu et al., 2025; Albalak et al., 2025). The other centers on refining RL algorithms themselves, including optimizations to PPO (Yuan et al., 2025b;a), extensions to GRPO variants (Yu et al., 2025; Liu et al., 2025; Zhang et al., 2025), entropy-regularized methods for exploration (Cui et al., 2025b; Yao et al., 2025; Wang et al., 2025), and alternative paradigms such as REINFORCE++ (Hu, 2025).

However, alongside this progress, a growing body of evidence has raised concerns about what these gains truly represent. Recent studies have shown that models trained with flawed or even adversarial reward signals can still achieve surprisingly strong results on standard mathematical bench-

marks (Shao et al., 2025). For example, rewards based purely on output format (e.g., the presence of expressions) can lead to improved scores regardless of correctness, and even models trained with random or inverted rewards have demonstrated non-trivial performance gains. These counterintuitive findings converge on a fundamental question: *Are RL-trained LLMs genuinely learning to reason, or are they merely exploiting superficial patterns embedded in benchmark datasets?* If benchmark success can be achieved without correctness, then current evaluation protocols may not be measuring true reasoning ability, which in turn calls into question the validity of benchmark-driven progress and highlights the need to reconsider what existing metrics actually assess.

At the core of this issue lies a structural limitation in how benchmarks are constructed. Most mathematical reasoning benchmarks present each problem as a single, fixed numerical instance. While this simplifies evaluation, it introduces two critical vulnerabilities. First, *benchmark contamination* is increasingly unavoidable. Many widely used datasets, such as AMC23 and AIME24&25, are sourced from public math competitions. Given the breadth of pretraining corpora, it is highly likely that some problems (or closely related variants) have appeared in training data, thereby confounding evaluations with memorization effects. Second, *evaluation instability* follows from the reliance on single-instance assessments. Since many competition-style math problems yield simple numeric answers (e.g., 0 or 1), models can often succeed through statistical priors, guesswork, or shallow heuristics rather than genuine reasoning. As a result, it becomes difficult to distinguish true problem-solving ability from superficial pattern exploitation.

To address these limitations, we propose **VAR-MATH**, a symbolic evaluation framework that probes true reasoning ability through *multi-instance verification*. As illustrated in Figure 1, the central idea is intuitive: *If a model genuinely understands a problem, it should solve not just one instance, but multiple variants that differ only in surface-level values while sharing the same underlying structure.* Concretely, VAR-MATH transforms fixed problems into symbolic templates by replacing constants with constrained variables. For example, the original question

$$\text{"Calculate the area defined by } ||x| - 1| + ||y| - 1| \leq 1\text{"}$$

can be generalized into

$$\text{"Calculate the area defined by } ||x| - a| + ||y| - a| \leq a\text{",}$$

where $a$ is sampled from a feasible domain. This symbolic multi-instantiation strategy shifts evaluation from one-shot correctness to structural consistency, thereby mitigating contamination, suppressing heuristic shortcuts, and enabling more faithful assessment of generalizable mathematical reasoning.

To quantify performance, VAR-MATH reports two complementary metrics. A *loose* score computes the average accuracy across sampled variants, while a *strict* score grants credit only if all variants of a problem are solved correctly. In addition, a bootstrapping procedure further stabilizes evaluation by reducing variance and yielding more reliable estimates.

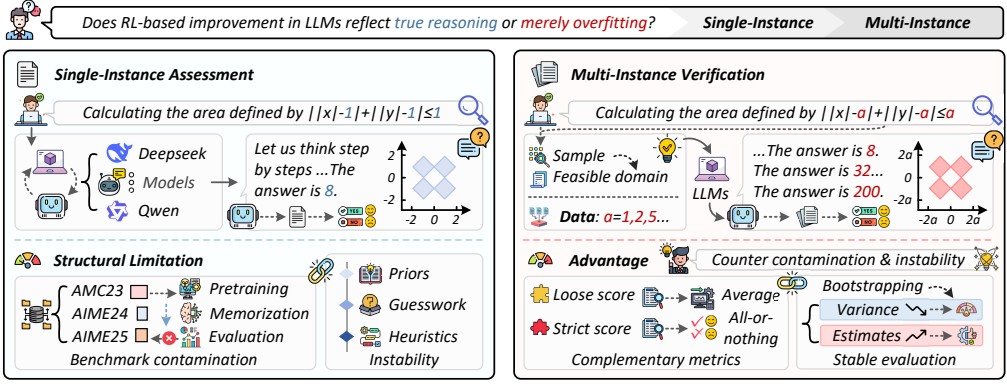

Figure 1: Multi-Instance Verification (VAR-MATH) vs. Single-Instance Assessment

Building on this protocol, we apply VAR-MATH to three widely used mathematical benchmarks, AMC23 (MAA, 2023), AIME24, and AIME25 (MAA), generating their symbolic counterparts VAR-AMC23, VAR-AIME24, and VAR-AIME25. When RL-finetuned models are re-evaluated on these transformed benchmarks, their performance drops sharply. For instance, several 7B-parameter models that previously achieved scores ranging from **36.9** to **78.6** on AMC23 drop to a range of **2.0** to **57.0** on VAR-AMC23, with similar declines on VAR-AIME24 and VAR-AIME25.

To summarize, our contributions are:

1. We propose VAR-MATH, a symbolic evaluation framework that systematically variabilizes three widely used benchmarks (AMC23, AIME24, and AIME25), enabling contamination-robust and consistency-based assessment of mathematical reasoning.

2. We establish a principled evaluation protocol that combines *loose* and *strict* consistency metrics with a bootstrapping procedure, ensuring statistically reliable comparison across models.

3. We conduct an extensive empirical study on VAR-AMC23, VAR-AIME24, and VAR-AIME25, revealing substantial performance declines in RL-finetuned models and exposing the limitations of current RL strategies in cultivating genuine reasoning.

## 2 RELATED WORK

A wide range of benchmarks has been developed to evaluate the mathematical reasoning capabilities of LLMs, spanning diverse difficulty levels, problem formats, and contamination risks. Existing efforts can be broadly categorized into static and dynamic benchmarks.

**Static Benchmarks** The GSM8K dataset (Cobbe et al., 2021) contains $8.5K$ grade-school math word problems ($7.5K$ train, $1K$ test) targeting multi-step arithmetic reasoning. While foundational, its limited numerical complexity reduces its utility for diagnosing advanced reasoning. MATH500 (Hendrycks et al., 2021) offers 500 high-school–level problems covering algebra and calculus. OlympiadBench (He et al., 2024) includes $8476$ Olympiad-level problems from sources such as the International Mathematical Olympiad and China's Gaokao, featuring multimodal inputs (e.g., diagrams) and step-by-step expert solutions for fine-grained evaluation in bilingual settings. AMC23 (MAA, 2023) collects problems from the 2023 American Mathematics Competition, emphasizing functional equations and complex analysis; each requires an integer answer between 0 and 999. Because of its small size and public availability, repeated sampling is necessary to reduce variance, while contamination remains a concern. The AIME series (MAA) is drawn from the American Invitational Mathematics Examination, with AIME24 containing 2024 contest problems and AIME25 adding novel problems curated in 2025. These increasingly challenging tasks demand deeper combinatorial and geometric reasoning, yet their public accessibility leaves them highly vulnerable to contamination as LLMs approach benchmark saturation.

**Dynamic Benchmarks** To mitigate contamination risks, recent work has shifted toward dynamic evaluation (Chen et al., 2025). For instance, Srivastava et al. (2024) alleviates contamination by creating functional variations of the MATH dataset, where new problems are generated by modifying numeric parameters to yield distinct solutions. Similarly, Mirzadeh et al. (2024) introduces an enhanced benchmark that generates diverse variants of GSM8K, while Gulati et al. (2024) alters variables, constants, and phrasing in Putnam competition problems. There are also several successful works that use symbolic variants (Xu et al., 2025; Li et al., 2024; Gao et al., 2022; Shi et al., 2023). LiveBench (White et al., 2024) further advances this direction by sourcing fresh problems monthly from arXiv papers, news, and contests, with rigorous contamination controls. However, maintaining such a benchmark requires ongoing human curation, limiting scalability and introducing subjectivity.

Overall, these studies represent important progress and highlight the value of functional variation in constructing dynamic benchmarks. Nevertheless, most existing efforts either concentrate on elementary problems (e.g., GSM8K, MATH) or remain restricted to Olympiad-style datasets. In contrast, our work focuses on advanced reasoning benchmarks that are widely adopted in RL-finetuning evaluations, including AMC23, AIME24, and AIME25. Building on this foundation, we introduce

VAR-MATH, a symbolic evaluation framework that not only variabilizes these benchmarks but also incorporates consistency-based metrics and bootstrapped stability analysis. This enables a more rigorous and reliable assessment of model reasoning robustness while further reducing susceptibility to data contamination.

# 3 VAR-MATH

As shown in Figure 2, VAR-MATH consists of three core components: its design principles, the data transformation process, and the evaluation protocol. We introduce each in turn below.

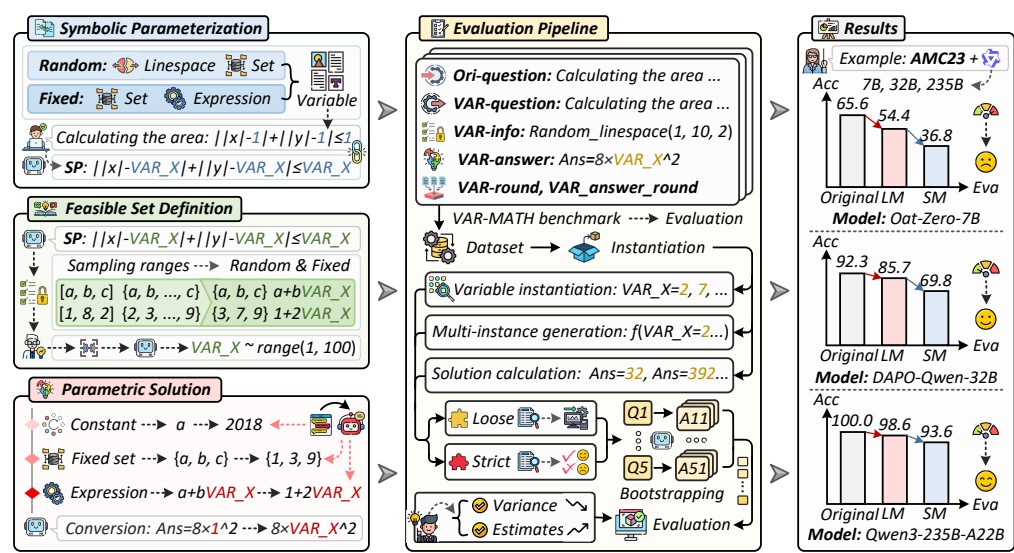

Figure 2: Overview of the VAR-MATH pipeline. The process consists of two stages: *preprocessing*, where original problems are symbolically abstracted by replacing constants with variables and defining feasible sampling ranges, and *evaluation*, where problems are instantiated into multiple concrete variants and assessed using loose (LM) and strict (SM) consistency metrics.

## 3.1 DESIGN PRINCIPLE

The core motivation behind VAR-MATH is to address two long-standing limitations in the evaluation of mathematical reasoning: *benchmark contamination* and *evaluation fragility*. Traditional benchmarks typically present problems as static numerical instances with fixed values, making them vulnerable to memorization and shallow pattern exploitation. In such settings, models may succeed by retrieving known solutions or leveraging statistical priors rather than performing genuine reasoning. These issues call for an evaluation paradigm that can separate true reasoning ability from superficial success.

VAR-MATH introduces such a paradigm through a process we call *symbolic variabilization*, which decouples problem structure from fixed numeric content. Instead of hardcoding specific constants, problems are restructured into symbolic templates, where concrete values are dynamically instantiated during evaluation. This abstraction allows models to be tested not on isolated instances, but across families of structurally equivalent problems.

The key assumption is that a model that truly understands a mathematical problem should demonstrate *reasoning consistency*, i.e., the ability to solve multiple variants of the same logical structure regardless of specific numerical values. By systematically sampling from constrained parameter spaces, VAR-MATH preserves the original semantics of each problem while introducing controlled variation. This results in a more robust and contamination-resistant evaluation protocol, which is capable of distinguishing genuine understanding from surface-level heuristics.

## 3.2 DATA PROCESSING

Building on the principle of symbolic variabilization, the data transformation pipeline systematically converts problems from established mathematical benchmarks into variabilized form. We focus on AMC23 and AIME24&25, which represent two distinct tiers of competition-level difficulty. Each selected problem undergoes symbolic abstraction through a structured four-step methodology:

- **Structural analysis.** Each problem is first solved independently by a mathematics expert. The expert works through the full reasoning process and cross-checks it against the official golden solution. This step identifies the algebraic structure of the problem, determines which quantities are essential constants, and isolates the core symbolic variables that drive the solution. We deliberately preserve the ratios or relationships that appear in the original derivation to maintain semantic fidelity.

- **Symbolic parameterization.** Key numerical constants are then replaced with symbolic variables. Feasible domains for each variable are chosen to stay close to the scale of the original values (e.g., an original value $x = 5$ may become a variable ranging from 2 to 8), while ensuring that the mathematical meaning of the problem remains valid. In constructing these domains, we apply domain restrictions from the derivation (such as positivity, non-vanishing denominators, or geometric constraints). Both continuous ranges and discrete sets are supported, as summarized in Table 1.

- **Parametric solution formulation and verification.** The final answer is expressed as a symbolic function of the newly defined variables. This symbolic formula is derived manually by the expert and then verified in two stages: (i) *human verification*: another annotator solves each instantiated variant directly from the rewritten prompt to ensure correctness; and (ii) *model verification*: we run all variants through a frontier model (DeepSeek) as an additional sanity check. A problem is accepted only if all of its variants are correctly solved during this step; otherwise, it will go through a second round of verification by another expert.

- **Variant sampling and evaluation protocol.** For each symbolic template, we uniformly sample values from the predefined feasible domains to generate a fixed set of up to $K = 5$ concrete variants (with $K = 2 \sim 4$ for a few problems to preserve difficulty alignment). All models are evaluated on *exactly* the same variants for a given problem, ensuring strict comparability across models. For each sampled variant, the ground-truth answer is computed directly from the parametric solution, and all instantiated variants are evaluated using a standardized prompting strategy consistent with prior mathematical reasoning benchmarks. This results in approximately 430 instantiated questions across the entire benchmark. An example is provided in Appendix G.

- **Precision specification.** To ensure numerical stability, we apply consistent rounding rules and significant-digit constraints to both the instantiated variables and the computed answers.

In certain cases, special constants integral to the mathematical identity of a problem (e.g., $\pi$, $e$, or fixed geometric parameters) are preserved without modification to maintain fidelity. The output of this pipeline is a set of variabilized benchmarks, namely **VAR-AMC23**, **VAR-AIME24**, and **VAR-AIME25**. Each problem is encoded as a structured object containing a symbolic expression, variable definitions with feasible sets, parametric answers, and metadata specifying its origin and difficulty. This unified representation enables efficient multi-instance instantiation and facilitates future benchmark extension and automation. Other details are provided in the Appendix.

## 4 EXPERIMENTS

### 4.1 EXPERIMENTAL SETUP

We evaluate model performance on six benchmarks: the original **AMC23**, **AIME24**, and **AIME25** datasets, together with their variabilized counterparts **VAR-AMC23**, **VAR-AIME24**, and **VAR-AIME25** generated by the VAR-MATH framework.

Table 1: Variable and Answer Expression Formats

| Variable Type (VAR_X) | Description |
|---|---|
| `Random_linespace_[a, b, c]` | Sampled from a linear space between $a$ and $b$ with $c$ intervals |
| `Random_Set_{a, b, ..., c}` | Sampled uniformly from the given discrete set |
| `Fixed_Set_{a, b, c}` | Must take one of the fixed values in the set |
| `Expression_a · VAR_Y + b` | Defined algebraically based on other variables |

| Answer Type | Description |
|---|---|
| `Constant a` | Answer is a constant value independent of input |
| `Fixed_Set_{a, b, c}` | Answer selected based on a fixed variable-to-output mapping |
| `Expression_a · VAR_Y + b` | Answer computed as a function of variable(s) |

**7B-parameter and 32B-parameter models.** We benchmark a collection of open-source 7B models, including the base `Qwen2.5-MATH-7B` (Yang et al., 2024), and several RL-enhanced variants: `Eurus-2-7B-PRIME` (Cui et al., 2025a), `Skywork-OR1-Math-7B` (He et al., 2025a), `SimpleRL-Zoo-7B` (Zeng et al., 2025), `Light-R1-7B-DS` (Wen et al., 2025), and `Oat-Zero-7B` (Liu et al., 2025). These models cover a range of RL training pipelines and policy optimization techniques. We further evaluate three 32B-scale models: the base `Qwen2.5-32B` (Team, 2024), and two RL-finetuned variants, `DAPO-Qwen-32B` (Yu et al., 2025) and `SRPO-Qwen-32B` (Zhang et al., 2025), both trained with large-scale reinforcement learning systems. Our evaluation pipeline is based on the open-source `Qwen2.5-MATH` repository[1], and employs **vLLM** (Kwon et al., 2023) for efficient decoding. All models are tested under consistent hardware and inference configurations on NVIDIA A6000 GPUs with `bfloat16` precision. Generation parameters are fixed at temperature = 0.6 and top-p = 1.0, while batch sizes are adjusted for each model to maximize throughput without affecting reproducibility.

**High-Capacity Models** We further include several high-capacity state-of-the-art models, including `DeepSeek-R1` (Guo et al., 2025), `SEED-THINK` (Seed et al., 2025), `Qwen3-235B-A22B` (Yang et al., 2025), and `OpenAI-o4-mini-high` (OpenAI, 2024). Evaluation for these models is conducted using a single inference pass per problem with default sampling configurations.

### 4.2 EVALUATION METRICS

We evaluate model performance using two complementary metrics: *loose* and *strict*. For each symbolic problem, up to five instantiated variants are generated by sampling values from the feasible domains of its parameters. Each variant is queried $M$ times, producing $M$ independent responses. The loose metric measures average correctness across all variants of a symbolic problem, while the strict metric enforces reasoning consistency: a problem is marked correct only if all of its variants are solved correctly. This all-or-nothing criterion emphasizes consistency across structurally equivalent problems rather than success on isolated instances.

To reduce variance and obtain statistically reliable estimates, we apply a bootstrap procedure with $N$ resampling rounds. For a given symbolic problem $Q$ with $K$ variants $\{Q_1, \ldots, Q_K\}$, and corresponding responses $\{A_{kj}\}$ for variant $Q_k$ under sample index $j = 1, \ldots, M$, the dataset is

$$D = \{(Q_k, A_{kj}) \mid k = 1, \ldots, K, \ j = 1, \ldots, M\}.$$

In each bootstrap round $i = 1, \ldots, N$, one response $\hat{A}_{ki}$ is drawn uniformly from $\{A_{kj}\}_{j=1}^{M}$ for every variant $Q_k$. The set $\{\hat{A}_{ki}\}_{k=1}^{K}$ is then used to compute both loose and strict scores:

$$\text{score}_{\text{loose}} = \frac{1}{N} \sum_{i=1}^{N} \left( \frac{1}{K} \sum_{k=1}^{K} \mathbf{1}[\hat{A}_{ki} = \text{gt}_k] \right), \quad (1)$$

$$\text{score}_{\text{strict}} = \frac{1}{N} \sum_{i=1}^{N} \left( \prod_{k=1}^{K} \mathbf{1}[\hat{A}_{ki} = \text{gt}_k] \right), \quad (2)$$

---

[1] `https://github.com/QwenLM/Qwen2.5-Math`

where $\mathrm{gt}_k$ denotes the ground-truth answer of $Q_k$. Final performance is reported as the mean of these bootstrap estimates, with standard deviations serving as a measure of statistical stability. An illustration of this procedure is provided in Appendix A.

## 4.3 MAIN RESULTS

Table 2: Evaluation Results on AMC23 and VAR-AMC23.

| Model | AMC23 | (strict) VAR-AMC23 | Drop | (loose) VAR-AMC23 | Drop |
|---|---|---|---|---|---|
| Qwen2.5-MATH-7B | 36.9 (6.3) | 2.0 (2.0) | -94.5% | 22.7 (2.6) | -38.5% |
| Eurus-2-7B-PRIME | 58.3 (4.3) | 28.9 (3.7) | -50.4% | 49.9 (2.5) | -14.3% |
| Skywork-OR1-Math-7B | 73.9 (5.4) | 57.0 (3.6) | -22.9% | 72.0 (2.3) | -2.6% |
| SimpleRL-Zoo-7B | 61.4 (4.8) | 33.6 (4.0) | -45.3% | 52.2 (2.3) | -15.0% |
| Light-R1-7B-DS | 78.6 (6.3) | 54.9 (4.6) | -30.2% | 75.8 (2.3) | -3.5% |
| Oat-Zero-7B | 65.6 (3.1) | 36.8 (3.3) | -43.9% | 54.4 (2.4) | -17.0% |
| Qwen2.5-32B | 33.4 (4.5) | 3.1 (2.5) | -90.6% | 27.4 (2.8) | -18.2% |
| DAPO-Qwen-32B | 92.3 (2.9) | 69.8 (3.1) | -24.4% | 85.7 (1.4) | -7.2% |
| SRPO-Qwen-32B | 86.7 (3.7) | 51.5 (4.5) | -40.6% | 73.9 (2.6) | -14.8% |
| DeepSeek-R1-0528 | 100.0 (0.0) | 96.4 (2.5) | -3.6% | 99.2 (0.5) | -0.8% |
| Qwen3-235B-A22B | 100.0 (0.0) | 93.6 (3.1) | -6.4% | 98.6 (0.7) | -1.4% |
| SEED-THINK-v1.6 | 100.0 (0.0) | 98.8 (1.5) | -1.2% | 99.8 (0.3) | -0.2% |
| OpenAI-o4-mini-high | 100.0 (0.0) | 93.4 (2.3) | -6.6% | 98.2 (0.7) | -1.8% |

### 4.3.1 RESULTS ANALYSIS ON THE STRICT METRIC

In this section, we focus on the strict metric, which emphasizes reasoning consistency across structurally equivalent problem variants, and we recommend it as the primary evaluation measure.

**RL-tuned 7B models show fragile generalization.** Across all benchmarks, RL-optimized 7B models experience sharp drops in accuracy once problems are variabilized. For example, `Light-R1-7B-DS` falls from 78.6 to 54.9 on AMC23, from 40.8 to 23.8 on AIME24, and from 32.7 to 17.1 on AIME25. Similar declines occur for `Eurus-2-7B-PRIME` and `Oat-Zero-7B`. These results point to two issues: overfitting to specific numeric templates, possibly amplified by contamination from public problem sets, and a lack of symbolic consistency, where solving one instance does not transfer reliably to others with altered values. Such weaknesses remain hidden under conventional single-instance benchmarks but are revealed by VAR-MATH.

**Scaling to 32B improves accuracy but inconsistency persists.** Larger 32B models achieve higher raw accuracy, e.g., `DAPO-Qwen-32B` and `SRPO-Qwen-32B` exceed 85 on AMC23. Nevertheless, they still suffer relative drops of more than 40% across the variabilized datasets, showing that scaling enhances memorization and structural recognition but does not fully resolve the problem of symbolic consistency.

**Frontier models are more robust yet still challenged by symbolic variation.** State-of-the-art models such as `DeepSeek-R1` and `SEED-THINK` maintain strong performance on AMC23, with drops below 5%. This robustness likely stems from high-quality training data and sophisticated alignment pipelines that mitigate shortcut learning. However, even these frontier systems experience notable degradation on the more difficult AIME24 and AIME25 variants, with relative drops up to 28.1%. These findings indicate that symbolic variation remains a fundamental challenge, underscoring the importance of evaluation protocols that move beyond surface-level accuracy toward consistency-based reasoning assessment.

The score drop mentioned above primarily results from data contamination and evaluation fragility. In the following text, we provide an in-depth analysis of data contamination and demonstrate how VAR-Math enhances assessment stability.

Table 3: Evaluation Results on AIME24 and VAR-AIME24.

| Model | AIME24 | (strict) VAR-AIME24 | Drop | (loose) VAR-AIME24 | Drop |
|---|---|---|---|---|---|
| Qwen2.5-MATH-7B | 10.8 (4.5) | 3.2 (2.7) | -70.0% | 7.9 (2.9) | -27.1% |
| Eurus-2-7B-PRIME | 15.8 (4.8) | 4.3 (2.9) | -72.5% | 13.4 (2.7) | -15.5% |
| Skywork-OR1-Math-7B | 41.5 (4.2) | 23.9 (4.3) | -42.3% | 39.0 (3.4) | -6.0% |
| SimpleRL-Zoo-7B | 23.8 (5.9) | 8.5 (3.7) | -64.1% | 20.4 (3.5) | -14.1% |
| Light-R1-7B-DS | 40.8 (5.1) | 23.8 (4.8) | -41.7% | 40.6 (3.3) | -0.6% |
| Oat-Zero-7B | 34.0 (2.1) | 12.8 (3.6) | -62.3% | 22.3 (2.5) | -34.3% |
| Qwen2.5-32B | 8.8 (4.4) | 2.3 (2.3) | -73.5% | 7.9 (2.6) | -9.6% |
| DAPO-Qwen-32B | 51.7 (6.6) | 29.8 (4.6) | -42.4% | 50.9 (2.8) | -1.5% |
| SRPO-Qwen-32B | 55.6 (5.0) | 29.2 (4.2) | -47.6% | 46.9 (2.9) | -15.7% |
| DeepSeek-R1-0528 | 86.8 (3.3) | 73.7 (3.8) | -15.1% | 82.3 (2.6) | -5.1% |
| Qwen3-235B-A22B | 84.1 (3.2) | 69.5 (3.4) | -17.4% | 80.1 (1.9) | -4.9% |
| SEED-THINK-v1.6 | 87.5 (3.3) | 73.4 (3.5) | -16.1% | 82.7 (2.4) | -5.5% |
| OpenAI-o4-mini-high | 91.8 (2.9) | 78.1 (3.7) | -14.9% | 89.0 (1.9) | -2.7% |

### 4.3.2 DECOUPLING THE IMPACT OF DATA CONTAMINATION

To better diagnose the sources of performance degradation, we analyze results under the *loose metric*. Unlike the strict all-or-nothing criterion, this softer metric grants partial credit for solving subsets of variants, thereby helping disentangle contamination-driven memorization from instability in symbolic reasoning.

Results on AMC23 suggest that contamination exerts a substantial influence, especially on the base models. For example, the base model Qwen2.5-MATH-7B shows a 38.5% decline, consistent with heavy reliance on memorized patterns rather than generalizable reasoning. By contrast, Skywork-OR1-Math-7B and DAPO-Qwen-32B record much smaller drops (2.6% and 7.2%, respectively), indicating greater resistance to contamination and stronger abstraction of underlying structures.

On the more challenging AIME24 and AIME25 benchmarks, degradation is more heterogeneous. Models such as SRPO-Qwen-32B exhibit relatively mild drops (e.g., 4.6% on AIME25), suggesting improved robustness across symbolic variants. Others, including Qwen2.5-32B and DAPO-Qwen-32B, suffer sharp declines (21.2% and 13.5% on AIME25), reflecting persistent fragility when faced with minor symbolic perturbations.

Together with the strict-metric results in Section 4.3.1, these findings point to two intertwined factors underlying symbolic degradation: benchmark-specific overfitting amplified by contamination, and instability in applying reasoning consistently across variants. While RL can improve scores on conventional benchmarks, it also risks reinforcing memorization and narrow heuristics. This underscores the necessity of evaluation frameworks that are both contamination-resistant and sensitive to reasoning stability.

### 4.3.3 ENHANCING EVALUATION STABILITY VIA VAR-MATH

In Appendix B, Figure 4 reports the distribution of standard deviations of the scores for 7B and 32B models on both the original and variabilized benchmarks. The result shows that VAR-MATH

Table 4: Evaluation Results on AIME25 and VAR-AIME25.

| Model | AIME25 | (strict) VAR-AIME25 | Drop | (loose) VAR-AIME25 | Drop |
|---|---|---|---|---|---|
| Qwen2.5-MATH-7B | 4.8 (3.1) | 0.0 (0.0) | -100.0% | 3.2 (1.3) | -34.2% |
| Eurus-2-7B-PRIME | 10.0 (3.1) | 1.2 (1.7) | -87.8% | 7.4 (1.4) | -26.0% |
| Skywork-OR1-Math-7B | 24.0 (3.8) | 15.0 (2.5) | -37.3% | 23.4 (1.6) | -2.4% |
| SimpleRL-Zoo-7B | 12.5 (3.4) | 2.9 (2.4) | -76.9% | 11.5 (1.6) | -7.9% |
| Light-R1-7B-DS | 32.7 (3.8) | 17.1 (3.2) | -47.7% | 30.3 (1.8) | -7.3% |
| Oat-Zero-7B | 9.2 (3.2) | 1.2 (1.8) | -87.4% | 8.4 (1.4) | -7.9% |
| Qwen2.5-32B | 3.5 (3.4) | 0.0 (0.0) | -100.0% | 2.8 (1.2) | -21.2% |
| DAPO-Qwen-32B | 37.3 (5.3) | 21.2 (2.4) | -43.2% | 32.2 (2.3) | -13.5% |
| SRPO-Qwen-32B | 26.5 (5.2) | 14.5 (3.0) | -45.2% | 25.2 (1.7) | -4.6% |
| DeepSeek-R1-0528 | 81.5 (3.3) | 61.3 (4.4) | -24.8% | 75.2 (2.5) | -7.9% |
| Qwen3-235B-A22B | 82.6 (3.1) | 61.6 (4.6) | -25.4% | 75.7 (2.0) | -8.1% |
| SEED-THINK-v1.6 | 81.7 (3.0) | 58.8 (4.0) | -28.1% | 75.3 (2.6) | -7.7% |
| OpenAI-o4-mini-high | 93.4 (2.4) | 76.7 (3.5) | -17.8% | 87.0 (1.9) | -6.8% |

consistently reduces output variance, with the effect most evident on the more challenging AIME25 benchmark, where conventional single-instance evaluation is highly susceptible to sampling noise.

This improvement derives from VAR-MATH's core design. By instantiating each symbolic problem multiple times and aggregating performance across variants, the framework dampens stochastic artifacts and outlier completions. Such ensemble-style averaging yields a more faithful estimate of reasoning ability and, with the incorporation of bootstrap methods to further stabilize estimates, provides a stable, interpretable signal of a model's true mathematical competence.

## 5 CONCLUSION

We introduced VAR-MATH, a systematic framework for evaluating mathematical reasoning in large language models. Targeting widely used benchmarks (AMC23, AIME24, and AIME25), VAR-MATH converts fixed problems into parameterized, multi-instance variants, enabling evaluation that is both contamination-resistant and consistency-based. To measure performance, we proposed complementary *loose* and *strict* metrics together with a bootstrap resampling procedure for stable statistical estimation.

Empirical results demonstrate that many RL-finetuned models, despite strong performance on conventional benchmarks, exhibit substantial degradation under VAR-MATH, exposing their reliance on dataset-specific artifacts and their limited generalization ability. These findings underscore the importance of principled dataset design and evaluation methodology in assessing reasoning competence. By enforcing consistency across variants and stabilizing measurement, VAR-MATH provides a more reliable indicator of genuine reasoning.

While our study centers on AMC23 and AIME24&25, the core methodology of VAR-MATH is broadly applicable. Future work includes extending this framework to richer mathematical domains and other reasoning-intensive tasks, such as program synthesis, formal logic, and decision-making. Such extensions hold promise for establishing more rigorous and generalizable evaluation standards. Another direction is to include strong SFT-only baselines (e.g., OpenThinker-3 Guha et al. (2025)) side-by-side to better quantify how much of the observed drop is due to RL-specific behavior versus general SFT math-tuning.

## ETHICS STATEMENT

We confirm that all authors of this submission have read and agree to abide by the ICLR Code of Ethics.

## REPRODUCIBILITY STATEMENT

The dataset and code are provided in the supplementary material to enable the reproduction of our results.

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

# A  BOOTSTRAP METHOD

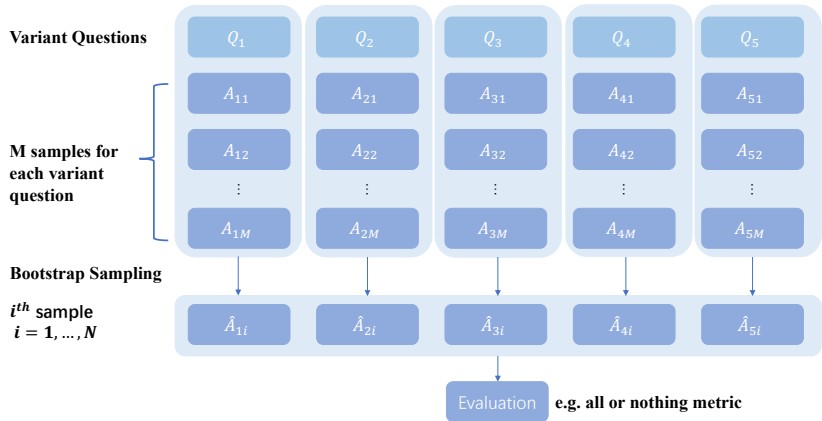

Figure 3: Illustration of the bootstrap procedure with $K = 5$ variants.

# B  ADDITIONAL EXPERIMENTAL RESULTS

Figure 4 reports the distribution of standard deviations in scores for 7B and 32B models, comparing the original benchmarks with their variabilized counterparts in the VAR-MATH suite. Across datasets, VAR-MATH consistently reduces variance, yielding more stable performance estimates. This effect is most pronounced on AIME25, where conventional single-instance evaluation is highly sensitive to sampling-induced fluctuations.

# C  DETAILS OF VAR-MATH CONSTRUCTION

Each problem in the VAR-MATH benchmark is represented as a structured object with the following fields:

1. *ori_question*: Original problem statement from the source dataset.

2. *ori_answer*: Corresponding reference (golden) answer.

3. *VAR_question*: Symbolic version of the problem, where numeric constants are abstracted into symbolic variables.

4. *VAR_info*: Definition of feasible sampling ranges for each symbolic variable.

5. *VAR_round*: Rounding precision (in significant digits) for computing numeric answers, implemented via `np.round` in Python.

6. *VAR_answer*: Symbolic expression of the answer as a function of abstract variables.

7. *VAR_answer_round*: Rounding precision applied to the final numerical output.

Representative examples are illustrated in Figure 5. We further show case studies for AMC23, AIME24, and AIME25 in Figures 6, 7, and 8, respectively.

# D  BENCHMARK STATISTICS

We summarize the detailed statistics of the VAR-MATH benchmark in Table 5.

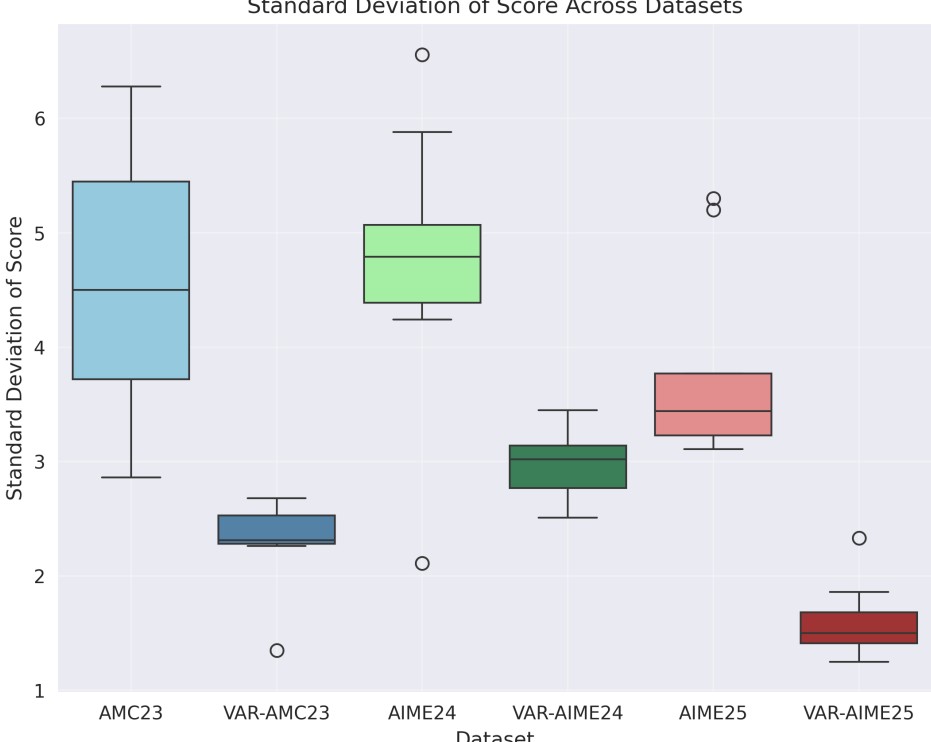

Figure 4: Standard deviation of model scores. VAR-MATH significantly reduces output variance across AMC23, AIME24, and AIME25.

Table 5: Statistics of the original and variabilized benchmark datasets.

| Dataset | Original Questions | Symbolizable Questions | Variant Questions |
|---|---|---|---|
| VAR-AMC23 | 40 | 37 | 183 |
| VAR-AIME24 | 30 | 24 | 126 |
| VAR-AIME25 | 30 | 25 | 130 |

# E  EVALUATION DETAILS

## E.1  DATASETS AND TESTING ENVIRONMENT

We evaluate model performance on six mathematical reasoning benchmarks: the original **AMC23**, **AIME24**, and **AIME25**, together with their variabilized counterparts **VAR-AMC23**, **VAR-AIME24**, and **VAR-AIME25**, constructed using the symbolic multi-instantiation pipeline described in Section 3. The original AMC23[2], AIME24[3], and AIME25[4] datasets are sourced from Hugging Face.

The evaluation framework is based on the open-source `Qwen2.5-MATH` repository[5], and uses PyTorch (v2.3.0), Transformers (v4.51.3), and vLLM (v0.5.1) for efficient decoding. All experiments are conducted on NVIDIA RTX A6000 GPUs with `bfloat16` precision.

---

[2] https://huggingface.co/datasets/zwhe99/amc23
[3] https://huggingface.co/datasets/Maxwell-Jia/AIME_2024
[4] https://huggingface.co/datasets/math-ai/aime25
[5] https://github.com/QwenLM/Qwen2.5-Math

| ori_answer | ori_question | VAR_question | VAR_info | VAR_round | VAR_answer | VAR_answer_round |
|---|---|---|---|---|---|---|
| 27 | Cities $A$ and $B$ are $45$ miles apart. Alicia lives in $A$ and Beth lives in $B$. Alicia bikes towards $B$ at 18 miles per hour. Leaving at the same time, Beth bikes toward $A$ at 12 miles per hour. How many miles from City $A$ will they be when they meet? | Cities $A$ and $B$ are $45$ miles apart. Alicia lives in $A$ and Beth lives in $B$. Alicia bikes towards $B$ at VAR_X miles per hour. Leaving at the same time, Beth bikes toward $A$ at VAR_Y miles per hour. How many miles from City $A$ will they be when they meet? | VAR_X=Random_linspace_[10,20,2] VAR_Y=Expression_30-VAR_X | 0 | Expression_45*VAR_X/ (VAR_X+VAR_Y) | 0 |
| 36 | Positive real numbers $x$ and $y$ satisfy $y^3=x^2$ and $(y-x)^2=4y^2$. What is $x+y$? | Positive real numbers $x$ and $y$ satisfy $y^3=x^2$ and $(y-x)^2=VAR_X y^2$. What is $x+y$? | VAR_X=Random_Set_{4,9,16,25,36} | 0 | Expression_(VAR_X**0.5+1)** 2*(VAR_X**0.5+2) | 0 |
| 45 | What is the degree measure of the acute angle formed by lines with slopes $2$ and $\frac{1}{3}$? | What is the degree measure of the acute angle formed by lines with slopes $VAR_Y$ and $VAR_X$? | VAR_X=Random_Set_{1/3,1/2,3/5} VAR_Y=Expression_(1+VAR_X) / (1-VAR_X) | -1 | Expression_45 | 0 |
| 3159 | What is the value of \[2^3 - 1^3 + 4^3 - 3^3 + 6^3 - 5^3 + \dots + 18^3 - 17^3?\] | What is the value of \[2^3 - 1^3 + 4^3 - 3^3 + 6^3 - 5^3 + \dots + VAR_X^3 - VAR_Y^3?\] | VAR_X=Random_linspace_[18,38,2] VAR_Y=Expression_VAR_X-1 | 0 | Expression_(VAR_X/2+1)*VAR _X*(VAR_X+1)-3/4*VAR_X**2 -VAR_X | 0 |
| 7 | How many complex numbers satisfy the equation $z^5=\overline{z}$, where $\overline{z}$ is the conjugate of the complex number $z$? | How many complex numbers satisfy the equation $z^VAR_X=\overline{z}$, where $\overline{z}$ is the conjugate of the complex number $z$? | VAR_X=Random_linspace_[5,25,2] | 0 | Expression_VAR_X+2 | 0 |
| 21 | Consider the set of complex numbers $z$ satisfying $\|1+z+z^{2}\|=4$. The maximum value of the imaginary part of $z$ can be written in the form $\frac{\sqrt{m}}{n}$, where $m$ and $n$ are relatively prime positive integers. What is $m+n$? | Consider the set of complex numbers $z$ satisfying $\|1+z+z^{2}\|=VAR_X$. The maximum value of the imaginary part of $z$ can be written in the form $\frac{\sqrt{m}}{n}$, where $m$ and $n$ are relatively prime positive integers. What is $m+n$? | VAR_X=Random_Set_{2,3,4,5,7} | 0 | Expression_5+VAR_X*4 | 0 |
| 3 | Flora the frog starts at 0 on the number line and makes a sequence of jumps to the right. In any one jump, independent of previous jumps, Flora leaps a positive integer distance $m$ with probability $\frac{1}{2^m}$. What is the probability that Flora will eventually land at 10? Write the answer as a simplified fraction $\frac{m}{n}$, find $m+n$ | Flora the frog starts at 0 on the number line and makes a sequence of jumps to the right. In any one jump, independent of previous jumps, Flora leaps a positive integer distance $m$ with probability $\frac{1}{2^m}$. What is the probability that Flora will eventually land at VAR_X? Write the answer as a simplified fraction $\frac{m}{n}$, find $m+n$ | VAR_X=Random_linspace_[10,200,13] | 0 | Expression_3 | 0 |
| 96 | Let $f$ be the unique function defined on the positive integers such that \[\sum_{d\mid n}d\cdot f\left(\frac{n}{d}\right)=1\] for all positive integers $n$. What is $f(2023)$? | Let $f$ be the unique function defined on the positive integers such that \[\sum_{d\mid n}d\cdot f\left(\frac{n}{d}\right)=1\] for all positive integers $n$. What is $f(VAR_X)$? | VAR_X=Fixed_Set_{2019,2021,2023,2027,2029} | 0 | d_Set_{1344,1932,96,-2026,-20 | 0 |
| 1 | How many ordered pairs of positive real numbers $(a,b)$ satisfy the equation \[(1+2a)(2+2b)(2a+b) = 32ab?\] | How many ordered pairs of positive real numbers $(a,b)$ satisfy the equation \[(1+VAR_Xa)(2+VAR_Yb)(VAR_Xa+VAR_Zb) = VAR_Uab?\] | VAR_X=Random_linspace_[1,10,1] VAR_Y=Random_linspace_[2,12,2] VAR_Z=Expression_int(VAR_Y/2) VAR_U=Expression_8*VAR_X*VAR_Y | 0 | Expression_1 | 0 |
| 8 | How many positive perfect squares less than $2023$ are divisible by $5$? | How many positive perfect squares less than $2023$ are divisible by $VAR_X$? | VAR_X=Random_Set_{3,5,7,11,13} | 0 | Expression_int(int(2023**0.5)/ VAR_X) | 0 |
| 18 | How many digits are in the base-ten representation of $8^5 \cdot 5^{10} \cdot 15^5$? | How many digits are in the base-ten representation of $8^VAR_X \cdot 5^{VAR_Y} \cdot 15^VAR_Z$? | VAR_Z=Random_Set_{3,4,5,6,7} VAR_X=Random_Set_{3,5,7,9,11} VAR_Y=Expression_3*VAR_X-VAR_Z | 0 | Expression_int(np.log(3)/ np.log(10)*VAR_Z)+1+3*VAR _X | 0 |

Figure 5: Illustrative examples of symbolic abstraction and metadata in VAR-MATH.

Question: Consider the set of complex numbers $z$ satisfying $|1 + z + z^2| = 4$. The maximum value of the imaginary part of $z$ can be written in the form $\frac{\sqrt{m}}{n}$, where $m$ and $n$ are relatively prime positive integers. What is $m + n$?

Answer: 21

Symbolic Question: Consider the set of complex numbers $z$ satisfying $|1 + z + z^2| = VAR\_X$. The maximum value of the imaginary part of $z$ can be written in the form $\frac{\sqrt{m}}{n}$, where $m$ and $n$ are relatively prime positive integers. What is $m + n$?

Feasible Set: VAR_X ~ $\{2,3,4,5, \dots \}$

Answer: $5 + VAR\_X * 4$

Figure 6: Example of original and symbolic variants from AMC23 and VAR-AMC23.

Question: Jen enters a lottery by picking 4 distinct numbers from $S = \{1,2,3, \cdots, 9,10\}$. 4 numbers are randomly chosen from $S$. She wins a prize if at least two of her numbers were 2 of the randomly chosen numbers, and wins the grand prize if all four of her numbers were the randomly chosen numbers. The probability of her winning the grand prize given that she won a prize is $\frac{\sqrt{m}}{n}$, where $m$ and $n$ are relatively prime positive integers. Find $m + n$.

Answer: 116

Symbolic Question: Jen enters a lottery by picking 4 distinct numbers from $S = \{1,2,3, \cdots, 9, VAR\_X\}$. 4 numbers are randomly chosen from $S$. She wins a prize if at least two of her numbers were 2 of the randomly chosen numbers, and wins the grand prize if all four of her numbers were the randomly chosen numbers. The probability of her winning the grand prize given that she won a prize is $\frac{\sqrt{m}}{n}$, where $m$ and $n$ are relatively prime positive integers. Find $m + n$.

Feasible Set: $VAR\_X \sim \{10,11, \dots, 20\}$

Answer: $(3 * VAR\_X - 11) * (VAR\_X - 4) + 2$

Figure 7: Example of original and symbolic variants from AIME24 and VAR-AIME24.

Question: Six points $A, B, C, D, E$ and $F$ lie in a straight line in that order. Suppose that $G$ is a point not on the line and that $AC = 26$, $BD = 22$, $CE = 31$, $DF = 33$, $AF = 73$, $CG = 40$, and $DG = 30$. Find the area of $\triangle BGE$.

Answer: 468

Symbolic Question: Six points $A, B, C, D, E$ and $F$ lie in a straight line in that order. Suppose that $G$ is a point not on the line and that $AC = VAR\_Y$, $BD = VAR\_Z$, $CE = 31$, $DF = 33$, $AF = VAR\_U$, $CG = 40$, and $DG = 30$. Find the area of $\triangle BGE$.

Feasible Set:
$VAR\_X \sim \{4, 5, \dots, 12\}$
$VAR\_Y = VAR\_X + 18$
$VAR\_Z = VAR\_X + 14$
$VAR\_U = VAR\_X + 65$

Answer: 12*(VAR_X+31)

Figure 8: Example of original and symbolic variants from AIME25 and VAR-AIME25.

### E.2 GENERATION CONFIGURATION

For 7B and 32B models, we adopt the system prompts and decoding configurations from their official implementations. The decoding hyperparameters are summarized in Table 6. High-capacity models are accessed via official APIs and evaluated using their default generation settings, without modification or additional prompts.

Table 6: Decoding and runtime configurations for model evaluation.

| Hyperparameter | Value |
|---|---|
| *General settings* | |
| Temperature | 0.6 |
| Number of generations | 16 |
| Top-$p$ | 1.0 |
| Use vLLM | True |
| GPU | NVIDIA RTX A6000 |
| *7B-parameter models* | |
| Max tokens per call | 8192 |
| GPUs used per model | 2 |
| M | 16 |
| N | 1000 |
| *32B-parameter models* | |
| Max tokens per call | 32768 |
| GPUs used per model | 4 |
| M | 16 |
| N | 1000 |
| *Frontier models* | |
| M | 4 |
| N | 1000 |

## F MORE DISCUSSION

### F.1 STRICT DROP APPLES-TO-ORANGES METRIC

A natural concern is that the performance drop between the original AMC/AIME datasets and their variabilized counterparts may partially arise from a mismatch in evaluation granularity: original scores are computed using pass@1, whereas strict VAR-AMC/AIME uses a consistency require-

ment across $K$ variants. This raises the question of whether the observed decline is an artifact of comparing pass@1 to a "$K/K$ strict" metric.

To address this, we introduce a *strict-AMC/AIME* metric that mirrors strict VAR-AMC/AIME. For each original problem, we perform $K$ independent inference runs and count the item as correct only if all $K$ runs are correct, using the same $K$, sampling strategy, and bootstrap procedure as in the variabilized evaluation. As shown in Tables 7–9, strict-AMC/AIME results remain close to the original pass@1 scores, whereas the drop from strict-AMC/AIME to strict-VAR-AMC/AIME remains large and consistent across models ($23\%/18\%/31\%$ on AMC, AIME24, and AIME25, respectively). This demonstrates that the strict consistency requirement itself does not account for the degradation.

Table 7: Strict-Metric Performance on AMC23 and VAR-AMC23. (Avg. Drop $-23.25\%$)

| Model | (strict) AMC23 | (strict) VAR-AMC23 | Drop |
|---|---|---|---|
| Qwen2.5-MATH-7B | 12.2 (4.0) | 2.0 (2.0) | -83.6% |
| Eurus-2-7B-PRIME | 40.8 (3.6) | 28.9 (3.7) | -29.2% |
| Skywork-OR1-Math-7B | 63.4 (3.2) | 57.0 (3.6) | -10.1% |
| SimpleRL-Zoo-7B | 42.1 (4.3) | 33.6 (4.0) | -20.2% |
| Light-R1-7B-DS | 59.9 (4.2) | 54.9 (4.6) | -8.3% |
| Oat-Zero-7B | 55.0 (3.1) | 36.8 (3.3) | -33.1% |
| Qwen2.5-32B | 6.5 (3.5) | 3.1 (2.5) | -52.3% |
| DAPO-Qwen-32B | 85.9 (3.4) | 69.8 (3.1) | -18.7% |
| SRPO-Qwen-32B | 72.4 (4.1) | 51.5 (4.5) | -28.9% |
| DeepSeek-R1-0528 | 100.0 (0.0) | 96.4 (2.5) | -3.6% |
| Qwen3-235B-A22B | 100.0 (0.0) | 93.6 (3.1) | -6.4% |
| SEED-THINK-v1.6 | 100.0 (0.0) | 98.8 (1.5) | -1.2% |
| OpenAI-o4-mini-high | 100.0 (0.0) | 93.4 (2.3) | -6.6% |

Table 8: Strict-Metric Performance on AIME24 and VAR-AIME24. (Avg. Drop $-17.87\%$)

| Model | (strict) AIME24 | (strict) VAR-AIME24 | Drop |
|---|---|---|---|
| Qwen2.5-MATH-7B | 3.4 (2.8) | 3.2 (2.7) | -5.9% |
| Eurus-2-7B-PRIME | 6.7 (2.6) | 4.3 (2.9) | -35.8% |
| Skywork-OR1-Math-7B | 27.1 (3.7) | 23.9 (4.3) | -11.8% |
| SimpleRL-Zoo-7B | 11.3 (4.1) | 8.5 (3.7) | -24.8% |
| Light-R1-7B-DS | 24.0 (4.7) | 23.8 (4.8) | -0.8% |
| Oat-Zero-7B | 25.1 (3.2) | 12.8 (3.6) | -49.0% |
| Qwen2.5-32B | 2.7 (2.4) | 2.3 (2.3) | -14.8% |
| DAPO-Qwen-32B | 36.0 (4.1) | 29.8 (4.6) | -17.2% |
| SRPO-Qwen-32B | 39.0 (4.5) | 29.2 (4.2) | -25.1% |
| DeepSeek-R1-0528 | 81.8 (2.8) | 73.7 (3.8) | -9.9% |
| Qwen3-235B-A22B | 80.8 (2.6) | 69.5 (3.4) | -14.0% |
| SEED-THINK-v1.6 | 82.8 (2.3) | 73.4 (3.5) | -11.4% |
| OpenAI-o4-mini-high | 88.5 (1.9) | 78.1 (3.7) | -11.8% |

Table 9: Strict-Metric Performance on AIME25 and VAR-AIME25. (Avg. Drop $-31.12\%$)

| Model | (strict) AIME25 | (strict) VAR-AIME25 | Drop |
|---|---|---|---|
| Qwen2.5-MATH-7B | 0.1 (0.5) | 0.0 (0.0) | -100.0% |
| Eurus-2-7B-PRIME | 1.9 (2.3) | 1.2 (1.7) | -36.8% |
| Skywork-OR1-Math-7B | 18.5 (1.9) | 15.0 (2.5) | -18.9% |
| SimpleRL-Zoo-7B | 3.8 (2.4) | 2.9 (2.4) | -23.7% |
| Light-R1-7B-DS | 22.1 (3.3) | 17.1 (3.2) | -22.6% |
| Oat-Zero-7B | 3.6 (2.3) | 1.2 (1.8) | -66.7% |
| Qwen2.5-32B | 0.0 (0.2) | 0.0 (0.0) | N/A |
| DAPO-Qwen-32B | 23.2 (3.6) | 21.2 (2.4) | -8.6% |
| SRPO-Qwen-32B | 17.0 (2.4) | 14.5 (3.0) | -14.7% |
| DeepSeek-R1-0528 | 76.6 (2.6) | 61.3 (4.4) | -20.0% |
| Qwen3-235B-A22B | 77.7 (1.6) | 61.6 (4.6) | -20.7% |
| SEED-THINK-v1.6 | 79.2 (2.5) | 58.8 (4.0) | -25.8% |
| OpenAI-o4-mini-high | 90.2 (0.8) | 76.7 (3.5) | -15.0% |

### F.2 STATISTICAL SIGNIFICANCE CHECK

To ensure that the observed differences are statistically reliable, we perform a one-sided $t$-test on the $M = 16$ independent inference runs for each open-weight model. For each model–benchmark pair, we test the null hypothesis

$$H_0 : \mu_{\text{loose}} = \mu_{\text{orig}},$$

i.e., the loose VAR-AMC/AIME score is equal to the original score, against the alternative hypothesis

$$H_1 : \mu_{\text{loose}} < \mu_{\text{orig}}.$$

This directly evaluates whether the variabilized versions lead to a statistically significant decline in performance under matched sampling conditions.

As shown in Tables 10–12, $18/27$ model–benchmark pairs yield $p < 0.05$, allowing us to reject $H_0$ with at least $95\%$ confidence. This confirms that, for the majority of settings, loose scores are significantly lower than original scores. For the remaining cases with $p \geq 0.05$, most correspond to models whose original accuracy is already very low (approximately $3.5 \sim 33$), leaving limited room for further decline and therefore a weaker statistical signal.

**Remark.** *We chose not to run the $t$-test directly on the bootstrap replicates because these are re-samples from the empirical distribution induced by the original $M = 16$ runs. Applying a $t$-test on the bootstrap draws would effectively compare two empirical distributions generated from the same finite sample, which in our experiments leads to uniformly tiny $p$-values (often $< 0.01$) and reflects the resampling procedure more than the underlying inference variability. To avoid overstating significance, we therefore conduct the $t$-test on the original $M$ independent runs, and use the bootstrap only to stabilize point estimates and confidence intervals.*

Table 10: Significance Check on AMC23 v.s. (loose) VAR-AMC23.

| Model | AMC23 | (loose) VAR-AMC23 | Drop | p-value |
|---|---|---|---|---|
| Qwen2.5-MATH-7B | 36.9 (6.3) | 22.7 (2.6) | -38.5% | $< 0.01$** |
| Eurus-2-7B-PRIME | 58.3 (4.3) | 49.9 (2.5) | -14.3% | $< 0.01$** |
| Skywork-OR1-Math-7B | 73.9 (5.4) | 72.0 (2.3) | -2.6% | 0.12 |
| SimpleRL-Zoo-7B | 61.4 (4.8) | 52.2 (2.3) | -15.0% | $< 0.01$** |
| Light-R1-7B-DS | 78.6 (6.3) | 75.8 (2.3) | -3.5% | 0.05* |
| Oat-Zero-7B | 65.6 (3.1) | 54.4 (2.4) | -17.0% | $< 0.01$** |
| Qwen2.5-32B | 33.4 (4.5) | 27.4 (2.8) | -18.2% | $< 0.01$** |
| DAPO-Qwen-32B | 92.3 (2.9) | 85.7 (1.4) | -7.2% | $< 0.01$** |
| SRPO-Qwen-32B | 86.7 (3.7) | 73.9 (2.6) | -14.8% | $< 0.01$** |

Table 11: Significance Check on AIME24 v.s. (loose) VAR-AIME24.

| Model | AIME24 | (loose) VAR-AIME24 | Drop | p-value |
|---|---|---|---|---|
| Qwen2.5-MATH-7B | 10.8 (4.5) | 7.9 (2.9) | -27.1% | 0.02* |
| Eurus-2-7B-PRIME | 15.8 (4.8) | 13.4 (2.7) | -15.5% | 0.04* |
| Skywork-OR1-Math-7B | 41.5 (4.2) | 39.0 (3.4) | -6.0% | 0.03* |
| SimpleRL-Zoo-7B | 23.8 (5.9) | 20.4 (3.5) | -14.1% | $< 0.01$** |
| Light-R1-7B-DS | 40.8 (5.1) | 40.6 (3.3) | -0.6% | 0.41 |
| Oat-Zero-7B | 34.0 (2.1) | 22.3 (2.5) | -34.3% | $< 0.01$** |
| Qwen2.5-32B | 8.8 (4.4) | 7.9 (2.6) | -9.6% | 0.27 |
| DAPO-Qwen-32B | 51.7 (6.6) | 50.9 (2.8) | -1.5% | 0.34 |
| SRPO-Qwen-32B | 55.6 (5.0) | 46.9 (2.9) | -15.7% | $< 0.01$** |

Table 12: Significance Check on AIME25 v.s. (loose) VAR-AIME25.

| Model | AIME25 | (loose) VAR-AIME25 | Drop | p-value |
|---|---|---|---|---|
| Qwen2.5-MATH-7B | 4.8 (3.1) | 3.2 (1.3) | -34.2% | 0.04* |
| Eurus-2-7B-PRIME | 10.0 (3.1) | 7.4 (1.4) | -26.0% | $< 0.01$** |
| Skywork-OR1-Math-7B | 24.0 (3.8) | 23.4 (1.6) | -2.4% | 0.31 |
| SimpleRL-Zoo-7B | 12.5 (3.4) | 11.5 (1.6) | -7.9% | 0.16 |
| Light-R1-7B-DS | 32.7 (3.8) | 30.3 (1.8) | -7.3% | 0.02* |
| Oat-Zero-7B | 9.2 (3.2) | 8.4 (1.4) | -7.9% | 0.24 |
| Qwen2.5-32B | 3.5 (3.4) | 2.8 (1.2) | -21.2% | 0.25 |
| DAPO-Qwen-32B | 37.3 (5.3) | 32.2 (2.3) | -13.5% | $< 0.01$** |
| SRPO-Qwen-32B | 26.5 (5.2) | 25.2 (1.7) | -4.6% | 0.18 |

## G  DETAILED EXAMPLE IN CONSTRUCTING THE PROBLEM

In this section, we provide two detailed examples (one easy and one hard) to illustrate the full construction pipeline.

---

**Example Conversion Pipeline (AMC 2023, Problem 11)**

**Original Problem.** What is the degree measure of the acute angle formed by lines with slopes 2 and $\frac{1}{3}$?

**1. Structural Analysis.** A mathematics expert solves the problem and identifies the key structure:

$$\tan(\theta) = \left| \frac{m_1 - m_2}{1 + m_1 m_2} \right|.$$

Essential constants are $m_1 = 2$ and $m_2 = \frac{1}{3}$, yielding $\theta = 45°$. This determines which quantities can be symbolized while preserving semantic fidelity.

**2. Symbolic Parameterization.** Key constants are replaced with variables:

$$m_1 = \text{VAR\_Y}, \qquad m_2 = \text{VAR\_X}.$$

Feasible domains:

$$\text{VAR\_X} \in \left\{ \tfrac{1}{3}, \tfrac{1}{2}, \tfrac{3}{5} \right\}, \qquad \text{VAR\_Y} = \frac{1 + \text{VAR\_X}}{1 - \text{VAR\_X}},$$

chosen to stay close to the original scale while ensuring validity (nonzero denominator, positive slopes) and similar difficulty (share the final solution step $\arctan 1 = 45°$).

**3. Parametric Solution Formulation.**

$$\tan(\theta) = \left| \frac{\text{VAR\_Y} - \text{VAR\_X}}{1 + \text{VAR\_Y} \cdot \text{VAR\_X}} \right| = 1 \quad \Rightarrow \quad \theta = 45°.$$

Thus the symbolic answer is:

$$\text{Answer} = 45°.$$

**4. Verification.**

- **Human verification:** another annotator solves each instantiated variant to confirm correctness and comparable difficulty.
- **Model verification:** all variants are run through a frontier model (DeepSeek). Any inconsistency triggers re-checking by an expert. Geometry problems are additionally validated using drawing tools.

**5. Variant Sampling and Evaluation.** Variants are instantiated by sampling from the feasible domains:

- Variant 1: $\text{VAR\_X} = \frac{1}{3}$, $\text{VAR\_Y} = 2$. Slopes: 2 and $\frac{1}{3}$. Ground truth: $45°$.
- Variant 2: $\text{VAR\_X} = \frac{1}{2}$, $\text{VAR\_Y} = 3$. Slopes: 3 and $\frac{1}{2}$. Ground truth: $45°$.
- Variant 3: $\text{VAR\_X} = \frac{3}{5}$, $\text{VAR\_Y} = 4$ (invalid). Slopes: 4 and $\frac{3}{5}$. Ground truth: $45°$.

Valid variants (typically $K = 3$ for this problem) are fixed and shared by all models. Loose/strict scores are computed using standardized prompting.

---

### Example Conversion Pipeline (AIME 2025 II, Problem 1)

**Original Problem.** Six points $A, B, C, D, E, F$ lie on a line in that order. A point $G$ is not on the line, and the distances satisfy

$$AC = 26, \quad BD = 22, \quad CE = 31, \quad DF = 33, \quad AF = 73, \quad CG = 40, \quad DG = 30.$$

Find the area of $\triangle BGE$.

**1. Structural Analysis.** Following the official solution, set

$$AB = a, \quad BC = b, \quad CD = c, \quad DE = d, \quad EF = e.$$

Then

$$a + b + c + d + e = AF = 73,$$
$$a + b = AC = 26,$$
$$b + c = BD = 22,$$
$$c + d = CE = 31,$$
$$d + e = DF = 33.$$

From these equations we deduce

$$c = 14, \qquad a + e = 34, \qquad b + c + d = 39.$$

Using Heron's formula on $\triangle CGD$ with side lengths $CG = 40$, $DG = 30$, and $CD = c = 14$ gives

$$[CGD] = \sqrt{42 \cdot 2 \cdot 12 \cdot 28} = 168.$$

Since $BE = b + c + d = 39$ and $CD = c = 14$ lie on the same line with the same altitude from $G$,

$$\frac{[BGE]}{[CGD]} = \frac{BE}{CD} = \frac{39}{14},$$

so

$$[BGE] = 168 \cdot \frac{39}{14} = 468.$$

This structure (solving for $c$ and $b + c + d$, then scaling areas by base ratio) is what we preserve in the symbolic version.

**2. Symbolic Parameterization.** We vary the lengths $AC$, $BD$, and $AF$ while keeping the configuration valid. Introduce a shift parameter

$$\text{VAR\_X} \in \{4, 5, 6, 7, 8, 9, 10, 11, 12\},$$

and define

$$AC = \text{VAR\_Y} = \text{VAR\_X} + 18,$$
$$BD = \text{VAR\_Z} = \text{VAR\_X} + 14,$$
$$AF = \text{VAR\_U} = \text{VAR\_X} + 65,$$

while keeping

$$CE = 31, \quad DF = 33, \quad CG = 40, \quad DG = 30$$

unchanged. The points remain ordered $A, B, C, D, E, F$ on the line and $G$ stays off the line.

**3. Parametric Solution Formulation.** With the same notation $AB = a$, $BC = b$, $CD = c$, $DE = d$, $EF = e$, the constraints become

$$a + b + c + d + e = AF = \text{VAR\_X} + 65,$$
$$a + b = AC = \text{VAR\_X} + 18,$$
$$b + c = BD = \text{VAR\_X} + 14,$$
$$c + d = CE = 31,$$
$$d + e = DF = 33.$$

From these equations we obtain, exactly as in the original case,

$$c = 14, \qquad d = 17, \qquad b = \text{VAR\_X}, \qquad b + c + d = \text{VAR\_X} + 31.$$

Thus $CD$ remains 14, so $\triangle CGD$ still has side lengths 40, 30, and 14, and its area is always

$$[CGD] = 168.$$

Meanwhile, the base of $\triangle BGE$ is

$$BE = b + c + d = \text{VAR\_X} + 31,$$

so with the same altitude from $G$,

$$\frac{[BGE]}{[CGD]} = \frac{BE}{CD} = \frac{\text{VAR\_X} + 31}{14},$$

which yields the parametric area

$$[BGE] = 168 \cdot \frac{\text{VAR\_X} + 31}{14} = 12(\text{VAR\_X} + 31).$$

Hence the symbolic answer is

$$\text{Answer} = 12(\text{VAR\_X} + 31).$$

**4. Verification.**

- **Human verification:** Another expert re-solves several instantiated variants using this symbolic derivation
- **Model verification:** All variants are also checked by a frontier model (DeepSeek) as a sanity check; any discrepancy triggers a second expert review to confirm correctness and comparable difficulty.

**5. Variant Sampling and Evaluation.** We sample VAR_X from its feasible set, for example:

- VAR_X = 4: area = $12(4 + 31) = 420$.
- VAR_X = 7: area = $12(7 + 31) = 456$.
- VAR_X = 11: area = $12(11 + 31) = 504$.
- ...

These variants are fixed and shared across all models. Loose/strict scores are computed using standardized prompting.

# H  THE USE OF LLMS

The authors utilized LLMs to assist with writing tasks such as text polishing and language refinement. All substantive intellectual content, research findings, and technical contributions are original to the authors. The LLM served only as a writing assistance tool under human supervision, and all output was critically evaluated and modified by the authors.

