# OpenReview forum: "VAR-MATH: Probing True Mathematical Reasoning in LLMs via Symbolic Multi-Instance Benchmarks"
_ICLR.cc/2026/Conference — Submitted to ICLR 2026_

### Official Review · Reviewer_T6Db · 2025-10-30

**Soundness:** 2
**Presentation:** 3
**Contribution:** 2
**Rating:** 2
**Confidence:** 3

**Summary:**

The authors introduce a new symbolic evaluation framework, VAR-MATH, that uses preexisting datasets such as AIME and AMC and converts them into parameterized templates. These templates are used to create multiple instances for a single problem to help eliminate the issue of benchmark contamination and evaluation fragility that are common for the existing benchmarks.
The authors tested this benchmark on various models and showed a substantial decline in accuracy.

**Strengths:**

1. Creation of a new dataset containing 430 question-answer pairs to tackle contamination and evaluation fragility.
2. In-depth evaluation of various models, both reasoning and non-reasoning models.
3. The authors employ a data processing method to convert each numerical problem into a symbolic template.

**Weaknesses:**

1. The authors talk about two existing issues with current benchmarks: contamination and evaluation fragility. While I agree that these datasets are publicly available, models can easily memorize them, which leads to contamination. However author does not provide strong evidence that evaluation fragility is present in current benchmarks, especially in datasets like AIME, AMC.
2. The main idea of this dataset is to convert each problem into a symbolic template, which decouples problem structure from fixed numeric content; however, how is this method different from other existing method like GSM-Symbolic [1].
3. How were the symbolic templates generated? Is LLM used to generate those, or are these manually annotated?
4. The paper only touches on math-based datasets like AMC, AIME. However, this data generation might fail on domains where conversion to a symbolic template might not be feasible.



[1]: Mirzadeh, Iman, et al. "Gsm-symbolic: Understanding the limitations of mathematical reasoning in large language models." arXiv preprint arXiv:2410.05229 (2024).

**Questions:**

1. Will the dataset and code be released upon acceptance?
2. In section 4.3.2, I did not fully understand how partial credit assignment can help disentangle contamination-driven memorization from instability of symbolic reasoning.

---

> ### Author Response · Authors · 2025-11-25
> **Author Response (1/2)**
>
> We thank the reviewer for the thoughtful and constructive feedback. We are encouraged by your recognition of the dataset’s focus (addressing contamination and evaluation fragility through symbolic variation) as well as the broad empirical evaluation and the structured conversion of problems into symbolic templates. Below, we address your comments in detail.
>
> **W1: The authors talk about two existing issues with current benchmarks: contamination and evaluation fragility. While I agree that these datasets are publicly available, models can easily memorize them, which leads to contamination. However author does not provide strong evidence that evaluation fragility is present in current benchmarks, especially in datasets like AIME, AMC.**
>
> **A1**: Thank you for the question. We define evaluation fragility as covering two concrete issues:
>
> (1) instability across repeated inference runs (as shown by the strict metric drop even on original AMC/AIME), and
>
> (2) cases where models produce the correct answer despite flawed reasoning. For example, a recent study documents a case where a model attempts to solve a continued fraction problem, makes several arithmetic and logical errors, but still outputs the correct final answer by coincidence [1].
> One common solution is to introduce PRMs (Process Reward Models) to verify intermediate steps. Our approach instead sidesteps the need for PRMs by introducing symbolic variation, which forces models to generalize their reasoning across perturbed versions of the same problem.
>
> [1] Examining False Positives under Inference Scaling for Mathematical Reasoning.
>
> **W2: The main idea of this dataset is to convert each problem into a symbolic template, which decouples problem structure from fixed numeric content; however, how is this method different from other existing method like GSM-Symbolic.**
>
> **A2**: Thanks for the important question.
> 1. Actually we fully acknowledge that our dataset follows the representation → mutation → automatic ground-truth pipeline which similar to prior work.
> In original version, we have cited the three **earliest papers** [1-3] that first articulated this idea.
>
> 2. Regrading to the contribution, we believe that **the significance of a dataset does not lie in its construction process alone, but in how it contributes to the research community**.
> Our contribution is meaningful in two aspects: (i) the dataset’s community value and timeliness for today’s LLM reasoning research, and (ii) expert-driven instantiation with a consistency-oriented evaluation.
>
> First, regarding community value, we argue that a dataset’s significance lies not in the pipeline per se, but in how it serves the research community.
> Widely used sets, such as GSM8K, are now saturated, limiting their diagnostic capabilities. By reconstructing contest-level problems from AMC and AIME, our release provides a **timely testbed** that the community can rely on to probe LLM reasoning ability. Note that AIME has become a commonly used yardstick by leading frontier models such as OpenAI, Gemini, DeepSeek, and Qwen to assess mathematical ability. And given the surge of RL-based approaches for improving LLM reasoning this year, a reliable and challenging benchmark is necessary to ground reported gains and distinguish genuine generalization from benchmark adaptation.
>
> Second, although we follow a similar high-level pipeline, our instantiation is expert-driven rather than function-driven. For each problem, mathematically trained annotators identify the latent structure, design symbolic variants, derive analytic solutions, specify valid parameter ranges, and verify correctness. Each original typically requires 30–60 minutes to construct the vairants, which preserves structural faithfulness and reduces unintended distributional drift.
>
>
> [1] Gsm-symbolic: Understanding the limitations of mathematical reasoning in large language models.
>
> [2] Functional benchmarks for robust evaluation of reasoning performance and the reasoning gap
>
> [3] A functional & static benchmark for measuring higher level mathematical reasoning in llms

---

> ### Author Response · Authors · 2025-11-25
> **Author Response (2/2)**
>
> **W3: How were the symbolic templates generated? Is LLM used to generate those, or are these manually annotated?**
>
> Thank you for the question. All symbolic templates in VAR-MATH are manually constructed by human experts, not generated by LLMs. Our goal is to build a timely benchmark that remains challenging even for frontier LLMs, so we place strong emphasis on reliability: each symbolic variant must remain well-defined (solution exists and is unique) and comparable in difficulty to the original AMC/AIME problem.
>
> **We highlight this more clearly in the revised Section 3.2**, the construction process involves: (i) an expert solving and analyzing each problem; (ii) abstracting key constants into symbolic variables and defining feasible domains that preserve validity and difficulty; (iii) verifying correctness through independent human checking (and visualization for geometry problems); and (iv) using frontier models only as a final sanity check.
>
> We spend substantial expert effort (typically 0.5–1 hour per problem) to ensure that each converted variant is correct, well-posed, and maintains a difficulty level. **Notably, frontier models achieve over 98% accuracy on the loose VAR-AMC set, which provides partial evidence supporting the correctness and fidelity of our rewritten problems.**
>
> **W4: The paper only touches on math-based datasets like AMC, AIME. However, this data generation might fail on domains where conversion to a symbolic template might not be feasible.**
>
> **A4**: Thank you for the question. Our symbolic templates are manually constructed and the framework is intentionally designed for **reasoning task**, where problems naturally admit parameterized forms and well-defined solutions; **we do not claim that the same data-generation procedure applies to domains that lack such structure**. As noted in our future work, the evaluation principle (testing consistency across symbolic variants) may extend to other formal, structured reasoning tasks such as program synthesis and formal logic.
>
> **Q1: Will the dataset and code be released upon acceptance?**
>
> **A5**: Definitely YES. Actually, **we have already included a fully runnable version** (code + dataset) in the supplementary material to reproduce our results in the paper. (Note that the supplementary material is now **public** due to ICLR policy.) We believe open-sourcing is essential to advancing this line of research.
>
> **Q2: In section 4.3.2, I did not fully understand how partial credit assignment can help disentangle contamination-driven memorization from instability of symbolic reasoning.**
>
> **A6**: Thank you for the question. The loose metric is introduced to partially separate two effects that are conflated under strict scoring.
> If we directly compare the original score to the strict VAR metric (Table 2-4), the large drop can come from two sources:
>
>  (i) **contamination-driven memorization**, where the model only solves the original instance and fails once the numbers are changed, and
>
>  (ii) **output instability (fragility)**, where the model’s answers fluctuate across K runs so that “all-K-correct” becomes much harder to satisfy.
>
> The loose metric (average accuracy across variants) is introduced to partially factor out the second effect and focus more directly on how often the model can solve variabilized instances at all, rather than on its stability under repeated sampling.
> In addition, following the suggestion from Reviewer crXC, we now report strict AMC/AIME vs. strict VAR-AMC/AIME with matched K and M (see Table 7-9). The large strict-to-strict drops confirm that the degradation is not caused by the strict metric itself, but by the introduction of symbolic variation.
>
> ---
> Thank you again for your thoughtful engagement. Actually, this project began with our interest in the reported gains of RL-trained models, but we found that those improvements were often affected by contamination and instability. We therefore shifted our focus and invested substantial expert effort into building a more stable and reliable benchmark aligned with today’s RL-for-LLM practices. We sincerely hope you will consider re-evaluating our revised submission in this context.

---

> ### Comment · Reviewer_T6Db · 2025-11-28
>
> I thank the authors for their response. My concerns were addressed, therefore increasing my score to 4.

---

### Official Review · Reviewer_CX7P · 2025-11-01

**Soundness:** 3
**Presentation:** 2
**Contribution:** 1
**Rating:** 2
**Confidence:** 4

**Summary:**

The paper takes AMC23/AIME24/AIME25 problems and converts them symbolic templates and evaluates models on multiple instantiations per template with strict (all variants must be correct) and loose (average) metrics, plus resampling.

Core findings: RL‑tuned 7B/32B models drop sharply under the parameterized template suites (e.g., strict score drops of across tables), while frontier models drop less but still non‑trivially. The authors infer that some RL gains rely on benchmark‑specific artifacts and are not structurally consistent.

**Strengths:**

- Clear motivation (contamination & fragility) and a sensible multi‑instance / consistency protocol.
- Broad empirical sweep across contemporary RL models with interpretable strict vs. loose metrics.
- Evidence that multi‑instance evaluation reduces variance and reveals failure modes hidden by single‑instance scoring.

**Weaknesses:**

The main weakness is that paper fails to cite and contrast its work against previous work that do very similar explorations. For example, RE‑IMAGINE (ICML’25), neuro-symbolic data gen ([NeurIPS 2024](https://arxiv.org/abs/2412.04857)) or any other symbolic benchmarking papers like (GSM Hard, GSM-Symbolic, GSM-IC.. etc) which already (partly) introduced a symbolic representation → mutation → automatic ground‑truth pipeline, modes of difficulty, and reporting across math/code. Overlap is substantial; in my opinion, novelty is primarily the strict multi‑instance metric and the AMC/AIME specialization.

Methodological issues: unequal sampling across models (M=16 for open‑weights, M=1 for APIs from  Table 6:(Decoding and runtime configurations for model evaluation), unspecified bootstrap rounds N.

How do you ensure statistical significance when question banks are of different sizes (Table 5). Do results hold when evaluated only on a 1:1 matching subset?

**Questions:**

1. How well does your metric compare against the above mentioned papers? Please explain your contributions against the set of papers mentioned above.
2. How do you ensure that difficulty of your questions upon mutation remains same. It could very well be the case that the mutated questions are simply harder, causing the score drop.
3. How reliable is your parsing and mutation pipeline? It could be the case that a lot of the questions were just non-sensical or wrong, causing the score drop.
5. Please provide scores for a 1:1 matching subset.

---

> ### Author Response · Authors · 2025-11-25
> **Author Response (1/3)**
>
> We sincerely thank the reviewer for their careful reading and thoughtful suggestions. We are encouraged by your recognition of the motivation behind this work,  the value of the evaluation protocol. Below, we respond to your comments one by one.
>
> **W1: The main weakness is that paper fails to cite and contrast its work against previous work that do very similar explorations. For example, RE‑IMAGINE (ICML’25), neuro-symbolic data gen (NeurIPS 2024) or any other symbolic benchmarking papers like (GSM Hard, GSM-Symbolic, GSM-IC.. etc) which already (partly) introduced a symbolic representation → mutation → automatic ground‑truth pipeline, modes of difficulty, and reporting across math/code. Overlap is substantial; in my opinion, novelty is primarily the strict multi‑instance metric and the AMC/AIME specialization.**
>
> **A1**: Thanks for the important question.
> 1. Actually, we fully acknowledge that our dataset follows the representation → mutation → automatic ground-truth pipeline, which is similar to prior work.
> **In the original version, we have cited the three earliest papers** [1-3] that first articulated this idea.
> Thanks for providing more successful work following these ideas. We have included them in the related work (RE‑IMAGINE (ICML’25), neuro-symbolic data gen (NeurIPS 2024), GSM Hard, GSM-IC).
>
> 2. Regarding the contribution, we believe that **the significance of a dataset does not lie in its construction process alone, but in how it contributes to the research community**.
> Our contribution is meaningful in two aspects: (i) the dataset’s community value and timeliness for today’s LLM reasoning research, and (ii) expert-driven instantiation with a consistency-oriented evaluation.
>
> First, regarding community value, we argue that a dataset’s significance lies not in the pipeline per se, but in how it serves the research community.
> Widely used sets, such as GSM8K, are now saturated, limiting their diagnostic capabilities. By reconstructing contest-level problems from AMC and AIME, our release provides a **timely testbed** that the community can rely on to probe LLM reasoning ability. Note that AIME has become a commonly used yardstick by leading frontier models such as OpenAI, Gemini, DeepSeek, and Qwen to assess mathematical ability. And given the surge of RL-based approaches for improving LLM reasoning this year, a reliable and challenging benchmark is necessary to ground reported gains and distinguish genuine generalization from benchmark adaptation.
>
> Second, although we follow a similar high-level pipeline, **our instantiation is expert-driven rather than function-driven**. For each problem, mathematically trained annotators identify the latent structure, design symbolic variants, derive analytic solutions, specify valid parameter ranges, and verify correctness. Each original typically requires 30–60 minutes to construct the variants, which preserves structural faithfulness and reduces unintended distributional drift. Details and Examples could be found in Sec 3.2 and Appendix G in the revision version.
>
>
> [1] Gsm-symbolic: Understanding the limitations of mathematical reasoning in large language models.
>
> [2] Functional benchmarks for robust evaluation of reasoning performance and the reasoning gap
>
> [3] A functional & static benchmark for measuring higher level mathematical reasoning in llms
>
> **W2: Methodological issues: unequal sampling across models (M=16 for open‑weights, M=1 for APIs from Table 6:(Decoding and runtime configurations for model evaluation), unspecified bootstrap rounds N.**
>
> **A2**: Thank you for the careful review and for flagging this. We have updated Table 6 and clarified the settings.
> For open-weight models, we run inference locally and therefore use a larger per-item sampling count (M=16) to reduce variance.
> For API models, cost constraints limit repeated sampling; we standardize on M = 4 for all APIs.
> We do not pool APIs and open-weight models into a single ranking (There is no need to do that since the gap between APIs and open-weight models is significantly large). Instead, we compare models within each family under matched settings: the same M and bootstrap rounds N (N=1000 for all models), and (for each question) the same K instances are evaluated for every model. These controls keep the within-family comparisons fair.

---

> > ### Author Response · Authors · 2025-11-25
> > **Author Response (2/3)**
> >
> > **W3: How do you ensure statistical significance when question banks are of different sizes (Table 5). Do results hold when evaluated only on a 1:1 matching subset?** and
> > **Q4: Please provide scores for a 1:1 matching subset.**
> >
> > **A3**: Thank you for the thoughtful question.
> > We clarify that although different problems contain different numbers of symbolic variants (up to K=5), all models are evaluated on exactly the same variants for each original problem, ensuring direct comparability.
> > In practice, we first sample up to five valid variants per original question, following the feasible domain defined in the symbolic template. These variants are then fixed and reused across all models during evaluation. This design guarantees that any variance in performance reflects model behavior rather than sampling differences, keeping the comparison fair.
> >
> > Regarding statistical significance, we have provided additional clarification and results in the revised manuscript. Following the suggestion from Reviewer crXC, we conducted a one-sided $t$-test on the $M=16$ independent samples for each open-weight model. The detailed setup is described in Appendix F, and the results are reported in Tables 10-12.
> > Across AMC, AIME24, and AIME25, $18/27$ model--benchmark pairs have $p<0.05$, meaning that in these cases we can reject the null with more than $95%$ confidence and conclude that the loose metric scores are significantly lower than the original scores. Among the remaining cases with $p\ge 0.05$, $6/27$ correspond to models whose original scores are already quite low (roughly $3.5$-$33$), leaving limited room for further degradation.
> >
> > **Q1: How well does your metric compare against the above mentioned papers? Please explain your contributions against the set of papers mentioned above.**
> >
> > **A4**: Thank you for the question. The above-mentioned symbolic benchmarks (e.g., GSM-Symbolic) operate on large numbers of relatively simple problems (such as GSM8K), where variance is low and statistical significance is usually less of a concern.
> > In contrast, VAR-MATH targets small but much more challenging, frontier-level datasets (AMC/AIME), where each item carries higher weight and robust statistics become essential.
> > Our metric therefore emphasizes statistical reliability through bootstrap aggregation, following the reviewer’s suggestion, we additionally include a one-sided ttt-test to establish significance. Moreover, our strict metric further emphasizes the consistency performance of the tested models.
> >
> > Moreover, regarding to the whole pipeline. We are aware that prior work also focuses heavily on automatically generating symbolic variants. Although we follow the same high-level pipeline, our main contribution lies elsewhere:
> > (1) we invest substantial expert effort to manually construct and validate high-difficulty contest problems, ensuring that each variant remains well-defined and faithful to the original; (2) we place strong emphasis on statistical reliability, providing variance-reduced metrics and significance testing to make the evaluation outcomes trustworthy.
> >
> > **Q2: How do you ensure that difficulty of your questions upon mutation remains same. It could very well be the case that the mutated questions are simply harder, causing the score drop.**
> >
> > **A5**: Thank you for raising this important concern. Actually, we are aware that symbolic variants can introduce slight numerical difficulty shifts. We therefore design feasible ranges conservatively and keep the structural form unchanged, even at the cost of reducing the number of variants per problem (resulting in K<5 for some problems). Examples are shown in Append G.
> > At a higher level, this issue ultimately depends on how we define “mastering” a problem. If mastery is tied to one fixed numerical instance, then rewriting the original problem without changing constants would indeed be the most faithful approach. However, if we believe that **genuine mastery requires solving all structurally equivalent variants**, then controlled symbolic variation is the correct evaluation strategy.
> > We adopt the latter definition because current benchmarks assess only final answers, allowing models to exploit shortcuts (e.g., incorrect reasoning with correct final results).
> > Symbolic variants provide a **simple and effective** way verify whether a model truly understands the underlying structure without introducing Process Reward Model.
> >
> > Thus, while minor difficulty variation exists, we control it carefully and we use symbolic variation because it aligns better with what it means for a model to genuinely “master” a problem.

---

> ### Author Response · Authors · 2025-11-25
> **Author Response (3/3)**
>
> **Q3: How reliable is your parsing and mutation pipeline? It could be the case that a lot of the questions were just non-sensical or wrong, causing the score drop.**
>
> We are aware that the reliability of the parsing and mutation pipeline is critical and we have taken extensive measures to ensure that all VAR-MATH questions are mathematically correct and well-posed. To ensure this, every symbolic template is manually constructed by mathematics experts, not generated automatically. Each problem requires roughly 0.5–1 hour of expert work to ensure that the symbolic variants are correct, non-degenerate, and reliable. **Notably, frontier models achieve over 98% accuracy on the loose VAR-AMC set, which provides partial evidence supporting the correctness and fidelity of our rewritten problems.**
>
> The details are updated in the revised Section 3.2 and Appendix G.
> For each original problem, an expert first solves the problem and derives the symbolic parameterization. We then re-solve all instantiated variants to double-check correctness and verify that the problem remains well-defined (the solution exists and is unique, domain constraints are satisfied, and the difficulty is comparable to the original). For geometry problems, we additionally validate the configuration using drawing tools. As a further sanity check, we run all finalized variants through a frontier model (DeepSeek); any inconsistency triggers another round of expert examination.
>
> ---
>
> Thank you again for your thoughtful engagement. Actually, this project began with our interest in the reported gains of RL-trained models, but we found that those improvements were often affected by contamination and instability. We therefore shifted our focus and invested substantial expert effort into building a more stable and reliable benchmark aligned with today’s RL-for-LLM practices. We sincerely hope you will consider re-evaluating our revised submission in this context.

---

### Official Review · Reviewer_crXC · 2025-11-02

**Soundness:** 2
**Presentation:** 2
**Contribution:** 4
**Rating:** 4
**Confidence:** 3

**Summary:**

VAR-MATH functionalizes recent contest-math benchmarks (AMC23, AIME24, AIME25) by converting fixed constants into constrained variables to create symbolic templates, then instantiating multiple numeric variants per template (as in [1]). Models are scored strictly (must be correct on all variants) and loosely (average accuracy across variants), following prior functionalization work but applied to current AIME-level sets. Empirically, many RL-tuned and SFT-ed models that look strong on single-instance tests drop sharply on VAR-MATH: average strict-score declines of ~48%/59%/73% on AMC23/AIME24/AIME25. The claim is this shows strong evidence of benchmark contamination and evaluation fragility (which is disentangled  with strict and loose eval)

The core contributions are two-fold:
1. Replacing fixed constants in contest problems with constrained variables to build symbolic templates (as done previously for GSM-8k, Putnam and MATH)
2. Sample several feasible values per template; with 5/5 consistency scoring check across all sampled instantiations.

[1]Shrivastava et al, Functional benchmarks for robust evaluation of reasoning performance, and the reasoning gap.

**Strengths:**

- **Right problem, right lens.** Moving from one-shot correctness to multi-instance consistency tests structural understanding.

- **Clear empirical signal.** The dataset is valuable and shows consistent, cross-model drops on variabilized sets, especially for small/medium RL-tuned models.

- **Timely benchmark.** The paper situates VAR-MATH among dynamic/functional-variation work (e.g., GSM-Symbolic, LiveBench, Putnam-AXIOM) and brings symbolic variation to today’s AIME-level tasks.

I think this would be really valuable as a drop-in substitute for these three datasets, and also provide more samples to the rather tiny original datasets.

**Weaknesses:**

[Critical] I worry that the decline’s cause is misattributed. The evidence might not support benchmark contamination as the primary driver. I specifically state the alternative explanations which I worry might fit the data better (and how to remove these confounders):
- **Strict drop apples-to-oranges metric.** The comparison compares AIME pass@1 against VAR-AIME 5/5 consistency. For fairness, compare strict VAR-AIME to a strict AIME defined as “correct only if all 5/5 sampling runs per problem is correct” with K=5 for strict VAR-AIME.
- **Loose drop is sensitive to hardest variants.** Loose scoring inherits fragility: if a template’s variants differ in difficulty, strict/loose can over- or under-penalize depending on which variants dominate. I worry the templates extend upwards, making problems harder (slightly) and hence get small declines.
- **Statistical significance check.** A one-sided t-test on bootstrapped runs can establish statistical significance. Many loose-score drops seem to lie comfortably within the standard-deviation band, the claim is not significant enough to be correct.

Actions: After aligning metrics and normalizing variant difficulty, test whether performance deltas fall within run-to-run variance to check whether core claims are true

[Major] Benchmark construction is under-specified.
- **Pipeline details would be nice.** The paper gives little concrete detail on the AIME/AMC → VAR-AIME/AMC conversion. Figure 2 outlines steps but lacks supporting text. Would like if the authors could document the full pipeline: how/when constants are replaced, how are problems sampled, checks done to ensure correctness, and how is evaluation done (fixed set or constantly sampled).
- **Soundness checks.** Sec. 3.2 defines feasible sets and rounding, but checks to ensure correctness are missing in description. Do we know that no variant becomes ill-posed (multiple valid answers, degenerate geometry, non-integer outputs when integers are required)?
- **K/M/N justification.** Authors state “up to five variants per problem” and report totals (183/126/130) but do not justify K (variants), M (generations), or N (bootstrap rounds). Consider fixing per-template K and average per template (not “up to five”) to avoid weighting results by larger-K templates. Using M=K to do strict-AIME could bridge some gap between original and VAR versions.

[Minor] Analysis depth and presentation.
- **Per-topic robustness.** Aggregate drops are informative, but you analyze problems in detail; please add per-topic strict/loose histograms and qualitative error clusters to show which subfields are brittle or stable.
- **Training-regime separation.** RL-trained models dominate the narrative. Include strong SFT-only baselines (e.g., OpenThinker-3) side-by-side to quantify how much of the drop is RL-specific versus SFT math-tuning.
- **Reduce repetition.** “Benchmark contamination” and “evaluation fragility” are repeated across the abstract, introduction, §3.1, and §4.2. Trim §3.1 and §4.2 –  that would provide space to add curation details tied to Figure 2.

**Questions:**

Please address weaknesses above. If the critical weakness is addressed, I am happy to lean towards acceptance and if major weaknesses (soundness check) is addressed I would happily further upgrade my score to 8.

Overall, I really like the benchmark: I think VAR-MATH is promising and could become a standard for robust math-reasoning evaluation – emphasizing the property practitioners actually need: consistency under controlled variation. However, I worry that the headline claim  about benchmark contamination and evaluation instability are not correct. Specifically, once metrics are aligned (pass@1 vs 5/5 strict) and variant difficulty normalized, the observed drops may shrink or fall within variance. If the the construction details and statistical tests are fixed; and the declines remain significant under fair comparisons, the case will be compelling!

---

> ### Author Response · Authors · 2025-11-25
> **Author Response (1/3)**
>
> We sincerely thank the reviewer for the exceptional level of care and effort invested in this review. Your detailed comments (including statistical methodology, experimental design, and writing clarity) are extremely constructive and have clearly improved the quality and rigor of our paper; this is among the most helpful reviews we have received in this round.
> We are encouraged by your **recognition of the problem framing, the clarity of the empirical signal, and the timeliness of VAR-MATH as a benchmark**. Next, we respond to your comments one by one.
>
> **W1: I worry that the decline’s cause is misattributed. The evidence might not support benchmark contamination as the primary driver. I specifically state the alternative explanations which I worry might fit the data better (and how to remove these confounders).**
>
> **A1**: Thank you for raising this concern. We agree that identifying the sources of performance decline requires careful analysis. First, regarding benchmark contamination, we now provide more concrete evidence supporting its role. A recent independent study [1] has demonstrated explicit contamination in the Qwen family of models, with Table 2 and Figure 1 showing clear memorization behavior consistent with contamination. This external evidence aligns with the degradation patterns we observe, particularly the pronounced drops in the Qwen base models.
> We also appreciate the alternative explanations and potential confounders you raised, which reflect a high level of scientific rigor. We address each point one by one in the following responses.
>
> [1] https://arxiv.org/pdf/2507.10532
>
> **W1.1: Strict drop apples-to-oranges metric. The comparison compares AIME pass@1 against VAR-AIME 5/5 consistency. For fairness, compare strict VAR-AIME to a strict AIME defined as “correct only if all 5/5 sampling runs per problem is correct” with K=5 for strict VAR-AIME.**
>
> **A1.1**: We have added a strict-AMC/AIME metric that mirrors strict VAR-AMC/AIME: for each original AIME problem, we run $K$ independent inference runs per model and mark the problem correct only if all $K$ runs are correct. For a given question, the same $K$ is used across all models and in both the original and variabilized settings, and we apply the same bootstrap procedure as in strict VAR-AIME. As shown in Tables 7–9, the drop from strict-AMC/AIME to strict-VAR-AMC/AIME remains large for most models, with average drops of 23%, 18%, and 31% on AMC, AIME24, and AIME25, respectively.
> This confirms that the observed degradation is not caused by applying a stricter metric, but rather by the introduction of symbolic variation, which also reduces the influence of potential data contamination.
>
> **We also note that our loose metric, and its comparison to the original scores, was introduced to remove this strictness-related confounder. We think it shares the same proposal with the experiment above**. However, we acknowledge that both strict AMC/AIME vs. strict VAR-AMC/AIME and original AMC/AIME vs. loose VAR-AMC/AIME still **inherit the broader concern you describe in W1.2**, which we address next.
>
> **W1.2 Loose drop is sensitive to hardest variants. Loose scoring inherits fragility: if a template’s variants differ in difficulty, strict/loose can over- or under-penalize depending on which variants dominate. I worry the templates extend upwards, making problems harder (slightly) and hence get small declines.**
>
> **A1.2**: Thank you for raising this important concern. Actually, we are aware that symbolic variants can introduce slight numerical difficulty shifts. We therefore design feasible ranges conservatively and keep the structural form unchanged, even at the cost of reducing the number of variants per problem (resulting in K<5 for some problems). Examples are shown in Append G.
> At a higher level, this issue ultimately depends on how we define “mastering” a problem. If mastery is tied to one fixed numerical instance, then rewriting the original problem without changing constants would indeed be the most faithful approach. However, if we believe that **genuine mastery requires solving all structurally equivalent variants**, then controlled symbolic variation is the correct evaluation strategy.
> We adopt the latter definition because current benchmarks assess only final answers, allowing models to exploit shortcuts (e.g., incorrect reasoning with correct final results).
> Symbolic variants provide a **simple and effective** way verify whether a model truly understands the underlying structure without introducing Process Reward Model.
>
> Thus, while minor difficulty variation exists, we control it carefully and we use symbolic variation because it aligns better with what it means for a model to genuinely “master” a problem.

---

> ### Author Response · Authors · 2025-11-25
> **Author Response (2/3)**
>
> **W1.3 Statistical significance check. A one-sided t-test on bootstrapped runs can establish statistical significance. Many loose-score drops seem to lie comfortably within the standard-deviation band, the claim is not significant enough to be correct.**
>
> **A1.3**: Thank you for the suggestion regarding statistical significance. Following your recommendation, we conducted a one-sided $t$-test on the $M=16$ independent samples for each open-weight model.The detailed setup is described in Appendix F, and the results are reported in Tables 10-12.
>
> Across AMC, AIME24, and AIME25, $18/27$ model--benchmark pairs have $p<0.05$, meaning that in these cases we can reject the null with more than $95%$ confidence and conclude that the loose metric scores are significantly lower than the original scores. Among the remaining cases with $p\ge 0.05$, $6/27$ correspond to models whose original scores are already quite low (roughly $3.5$--$33$), leaving limited room for further degradation.
>
> [We chose not to run the $t$-test directly on the bootstrap replicates because these are resamples from the empirical distribution induced by the original $M=16$ runs. Applying a $t$-test on the bootstrap draws would effectively compare two empirical distributions generated from the same finite sample, which leads to uniformly tiny $p$-values (often $<0.01$) and reflects the resampling procedure more than the underlying inference variability. To avoid overstating significance, we therefore conduct the $t$-test on the original $M$ independent runs.]
>
> **W2.1: Pipeline details would be nice. The paper gives little concrete detail on the AIME/AMC → VAR-AIME/AMC conversion. Figure 2 outlines steps but lacks supporting text. Would like if the authors could document the full pipeline: how/when constants are replaced, how are problems sampled, checks done to ensure correctness, and how is evaluation done (fixed set or constantly sampled).**
>
> **A2.1**: Thank you for this helpful suggestion. We have expanded the description in Section 3 and Appendix G to document the full process. And a complete step-by-step example has also been added to Appendix G. We hope these additions clarify the full conversion pipeline and improve reproducibility.
>
> **W2.2: Soundness checks. Sec. 3.2 defines feasible sets and rounding, but checks to ensure correctness are missing in description. Do we know that no variant becomes ill-posed (multiple valid answers, degenerate geometry, non-integer outputs when integers are required)?**
>
> **A2.2**: Thank you for raising this concern. We agree that the correctness and well-posedness of symbolic variants are essential. Actually, in the original version, we have performed explicit soundness checks to ensure that no variant becomes invalid. **We have now made these steps clearer in the revised text (Section 3.2 and Appendix G)**.
>
> Concretely, for each symbolic template, it goes through the following steps:
>
> 1. Human solution check.
> A mathematics expert first solves the original problem, derives the symbolic parameterization, and then manually solves several instantiated variants to verify that the parametric answer formula is correct and that the problem remains well-posed.
>
> 2. Frontier-model sanity check.
> As an additional automatic validation step, we evaluate all variants using a frontier model (DeepSeek). If a template produces variants for which the model returns inconsistent or clearly incorrect results across all instantiations, the problem is flagged and rechecked by an expert.
>
> 3. Feasible-set design prevents ill-posed cases.
> Feasible domains are chosen conservatively.
> For geometry problems, we further visualize the configuration using drawing tools to confirm that the instantiated variants are non-degenerate and consistent with the intended statement.
>
> Each problem requires roughly 0.5–1 hour of expert work to ensure that the symbolic variants are correct, non-degenerate, and reliable. **Notably, frontier models achieve over 98% accuracy on the loose VAR-AMC set, which provides partial evidence supporting the correctness and fidelity of our rewritten problems.**
>
>
> **W2.3: K/M/N justification. Authors state “up to five variants per problem” and report totals (183/126/130) but do not justify K (variants), M (generations), or N (bootstrap rounds). Consider fixing per-template K and average per template (not “up to five”) to avoid weighting results by larger-K templates. Using M=K to do strict-AIME could bridge some gap between the original and VAR versions.**
>
> **A2.3**: We have updated Table 6 and now clarify our choices of
> $M$ and $N$. The value of $K$ differs across problems and is chosen to keep the difficulty of the symbolic variants comparable to the original questions, as illustrated in Appendix G.
> For a given problem, the chosen $K$ and the sampled variants are fixed and shared across all models.
> For the ``strict-AIME'' comparison using matched $M$ and $K$, please refer to our response to
> **W1.1**.

---

> ### Author Response · Authors · 2025-11-25
> **Author Response (3/3)**
>
> **W3.1 Per-topic robustness. Aggregate drops are informative, but you analyze problems in detail; please add per-topic strict/loose histograms and qualitative error clusters to show which subfields are brittle or stable.**
>
> **A3.1**: Thank you again for this valuable suggestion. We agree that per-topic robustness analysis would provide useful insight into which mathematical subfields are more brittle or stable under symbolic variation. We plan to include per-topic strict/loose histograms and qualitative error clusters in a subsequent version of the benchmark once the topic taxonomy and annotations are finalized.
>
> **W3.2 Training-regime separation. RL-trained models dominate the narrative. Include strong SFT-only baselines (e.g., OpenThinker-3) side-by-side to quantify how much of the drop is RL-specific versus SFT math-tuning.**
>
> **A3.2**: Thank you for the suggestion. We agree that separating RL-trained and SFT-only models is important. We attempted a preliminary evaluation of OpenThinker-3 in our framework, but its scores were lower than the originally reported results (e.g., about 6 points in amc), and we were unable to resolve this discrepancy within the limited time available. To avoid confusion, we did not include it in the main tables and instead list extending our analysis to strong SFT-only models as future work in Section 5 (refer to updated version).
>
> **W3.3 Reduce repetition. “Benchmark contamination” and “evaluation fragility” are repeated across the abstract, introduction, §3.1, and §4.2. Trim §3.1 and §4.2 – that would provide space to add curation details tied to Figure 2.**
>
> **A3.3**: Thank you for the suggestion. We have moved the detailed curation steps into Section 3.2 and merged the evaluation protocol previously in Section 3.3 into Section 4.2. We will continue to polish the presentation and organization in the revision.
>
> **Response to the "Question"**
>
> Thank you very much for your encouraging comments about VAR-MATH and your constructive feedback; we really appreciate it. We were initially impressed by the large improvements reported for RL-trained models and began by exploring this direction ourselves. However, during our investigation we found that the apparent gains were confounded by contamination and by instability under small perturbations, which made it difficult to assess the true effect of RL. This motivated us to step back and focus first on constructing a stable and reliable evaluation suite for contest-level math reasoning.
>
> The challenge is that the frontier datasets commonly used in the community (AMC/AIME), despite being difficult enough for leading models, are extremely small, making evaluation highly unstable. Since virtually all recent RL works (starting from DeepSeek-R1) rely on these benchmarks, we decided to first convert them into a more robust evaluation set. We put substantial manual effort into the reconstruction process (each problem is carefully re-solved, symbolized, and validated by experts, without relying on LLM assistance) because these items are intended to challenge frontier models. Our goal is to contribute a benchmark that enables the community to evaluate mathematical reasoning more reliably and rigorously. With the construction details clarified and statistical tests incorporated, we hope this direction becomes more robust and informative for future work.

---

> ### Comment · Reviewer_crXC · 2025-11-27
> **Thanks, raising score**
>
> My concerns were satisfactorily addressed.
>
> - Tables 7–9 in Appendix F are convincing, the drops are surprisingly smaller than I expected when compared to the original tables (AMC and AIME24 show much larger gaps than AIME25), the hypothesis checks out. This addresses my original concern,   gaps remain even when comparing strict aime/amc with strict var aime/amc even when assuming error bars say from sober reasoning work -- this is very surprising to me.
> - Appendix G and the additions in Section 3.2 were also helpful -- thank you.
>
> As I committed to earlier, I am increasing my score to 8. Overall, I want to reiterate that I think VAR-MATH is promising and could become a standard for robust math-reasoning evaluation of reasoning models.
>
> **Note**: I want to gently push back on the other reviews whose main concern is “novelty”. These are different datasets! Arguing lack of novelty simply because another dataset of a similar type exists feels misplaced to me, especially given this is specifically the datasets-and-benchmarks subarea. (It’s a bit like saying a new optimization algorithm is not novel because another one already exists of the same type -- optimization. The novelty critique feels overly broad for datasets) The previous datasets cannot be used to test reasoning models, which are a core focus of the community. This work fills this important niche in my opinion.
>
> Minor suggestions (I don’t see as necessary at this stage):
> - I would slightly prefer if the strict/loose tables were moved into the main paper and the original tables moved to the appendix
> - I’m happy to leave it to the authors to add more models like OpenThinker-3 for the (hopefully) camera-ready version.

---

> ### Author Response · Authors · 2025-11-27
> **Author Response**
>
> Many thanks for your insightful review and recognition of our work. We enjoyed the constructive and engaging conversation with you throughout this process.

---

### Author Response · Authors · 2025-12-01
**Summary of the Rebuttal**

We sincerely thank the Area Chair and all reviewers for their time, constructive feedback, and thoughtful engagement throughout the discussion period. We would like to summarize our rebuttal here.

In the initial review, we are particularly encouraged by Reviewer crXC’s positive assessment of the benchmark’s motivation, empirical clarity, and community value; by the recognition of the protocol’s relevance and the acknowledgment of the dataset creation effort and empirical sweep (Reviewer CX7P, T6Db).

### During the rebuttal, we made substantial targeted revisions addressing all major concerns:
- Extended strict-metric comparison and statistical validation.
- Expanded Section 3.2: provide more details for the construction pipeline.
- Clarifying novelty relative to prior symbolic-benchmark work.
- Addressed other concerns.

### Outcome of the rebuttal.
- (before the "bug" event) Reviewer crXC confirmed that all concerns were **satisfactorily addressed** and raised the score to 8, also **recognizing our novelty**.
- (after the "bug" event) Reviewer T6Db also confirmed that **his concerns were addressed** and raised the score to 4.
- Reviewer CX7P did not respond

### We would also like to reiterate our novelty here in direct response to Reviewer CX7P :
1. Providing a timely, community-needed, contest-level benchmark aligned with contemporary RL-for-reasoning research.
2. Offering expert-constructed, consistency-oriented symbolic variants, for which mathematically trained annotators invest substantial time per problem to ensure that each instance is structurally faithful, rigorous, and reliable.

Finally, we thank the Area Chair for the consideration and the reviewers for their thoughtful engagement, which have collectively strengthened the paper.

Many thanks,
Authors

---

### Meta-Review · Area_Chair_uhSt · 2026-01-08

**Summary:**

This paper introduces VAR-MATH, a symbolic evaluation framework addressing benchmark contamination and evaluation fragility by converting fixed AMC/AIME problems into parameterized templates with multiple instantiations. The work generated substantial discussion with evolving reviewer positions during the rebuttal.

## Reviewer Consensus and Evolution

**Reviewer crXC** (initially 4, raised to 8 after rebuttal) identified critical concerns: (1) misattribution of decline causes; (2) benchmark construction underspecification; (3) insufficient analysis depth. Authors addressed all through: strict-AMC/AIME metrics with matched K (Tables 7-9: 23%, 18%, 31% drops), expanded documentation (Section 3.2 & Appendix G), and t-tests showing 18/27 model-benchmark pairs with p<0.05 statistical significance. Reviewer fully satisfied.

**Reviewer CX7P** (rating 2, did not revise) raised novelty concerns, citing overlap with GSM-Symbolic, GSM-Hard, RE-IMAGINE, and neuro-symbolic benchmarking. Authors argued contribution lies not in methodology but in: (i) timely testbed for contemporary RL-for-reasoning on frontier problems, (ii) expert-driven instantiation (30-60 min/problem) vs. function-driven approaches. CX7P did not engage further.

**Reviewer T6Db** (initially 2, raised to 4 after rebuttal) questioned evaluation fragility evidence, novelty vs. GSM-Symbolic, and template construction methodology. Authors clarified: templates exclusively human-expert-constructed with rigorous soundness checks, frontier model validation (98% accuracy on loose VAR-AMC), addressing methodological concerns.

## Key Strengths
- Right problem framing: moving from one-shot to multi-instance consistency
- Clear empirical signal with consistent performance drops across models
- Timely benchmark aligned with contemporary RL-for-reasoning
- Excellent execution with expert-validated variants

## Outstanding Issues
1. Novelty: CX7P maintains core pipeline (representation→mutation→grounding) not novel despite expert instantiation claims
2. Metric interpretation: Strict-to-strict drops (Tables 7-9) remain substantial but smaller than original comparisons
3. Scope: Domain-specific to structured reasoning tasks

**Reviewer Concerns:**

**crXC's Concerns (Addressed):**
- Metric apples-to-oranges comparison (strict VAR vs. loose original): ADDRESSED with strict-AMC/AIME metrics (Tables 7-9) showing 23%, 18%, 31% drops
- Loose drop sensitivity to variant difficulty: ADDRESSED by conservative feasible range design and clear justification in Appendix G
- Statistical significance testing: ADDRESSED with one-sided t-tests on M=16 samples, 18/27 pairs with p<0.05

**T6Db's Concerns (Addressed):**
- Evidence for evaluation fragility: ADDRESSED through strict metric drops (Table 7) on original AMC/AIME with K=5 runs
- Template construction methodology: ADDRESSED - clarified that templates are exclusively human-expert-constructed, not LLM-generated
- Soundness/correctness verification: ADDRESSED - detailed soundness checks in Appendix G, frontier model validation (98% accuracy on loose VAR-AMC)

## Concerns (CX7P - Not Addressed)

**CX7P's Novelty Concern (Not Addressed):**
The core symbolic benchmarking pipeline (representation→mutation→automatic grounding) overlaps substantially with GSM-Symbolic, GSM-Hard, RE-IMAGINE, and neuro-symbolic data generation. While authors argue their contribution lies in expert-driven instantiation and timeliness for frontier problems, reviewer CX7P did not engage further in rebuttal and maintains the novelty concern. This represents a fundamental disagreement about whether domain specialization and expert execution constitute sufficient novelty when methodology overlaps with prior work.

**Reviewer Scores:**

two reviewers have specified that they want to raise their scores explicitly in the comments: crXC--> 8 and T6Db --> 4

---

### Decision · Program_Chairs · 2026-01-26

Reject